# Historical (1750 – 2014) anthropogenic emissions of reactive gases and aerosols from the Community Emission Data System (CEDS)

Rachel M. Hoesly[1], Steven J. Smith[1,2], Leyang Feng[1], Zbigniew Klimont[3], Greet Janssens-Maenhout[4], Tyler Pitkanen[1], Jonathan J. Seibert[1], Linh Vu[1], Robert J. Andres[5], Ryan M. Bolt[1], Tami C. Bond[6], Laura Dawidowski[7], Nazar Kholod[1], Jun-ichi Kurokawa[8], Meng Li[9], Liang Liu[6], Zifeng Lu[10], Maria Cecilia P. Moura[1],  Patrick R. O'Rourke[1], Qiang Zhang[9]

[1] Joint Global Change Research Institute, Pacific Northwest National Lab, College Park, MD, 20740 USA
[2] Department of Atmospheric and Oceanic Science, University of Maryland, College Park, Maryland, 20742 USA
[3] International Institute for Applied Systems Analysis, Laxenburg, Austria
[4] European Commission, Joint Research Centre, Directorate Energy, Transport & Climate, Via Fermi 2749, I-21027 ISPRA, Italy
[5] Carbon Dioxide Information Analysis Center, Oak Ridge National Laboratory, Oak Ridge, TN 37831-6290 USA
[6] Dept. of Civil & Environmental Engineering, University of Illinois at Urbana-Champaign, Urbana, IL, 61801 USA
[7] Comisión Nacional de Energía Atómica, Buenos Aires, Argentina
[8] Japan Environmental Sanitation Center, Asia Center for Air Pollution Research, Atmospheric Research Department
[9] Department of Earth System Science, Tsinghua University, Beijing, China
[10] Energy Systems Division, Argonne National Laboratory, Argonne, IL, USA

*Correspondence to*: Rachel M. Hoesly (rachel.hoesly@pnnl.gov) and Steven J. Smith (ssmith@pnnl.gov)

**Abstract.** We present a new data set of annual historical (1750 - 2014) anthropogenic chemically reactive gases (CO, $CH_4$, $NH_3$, $NO_X$, $SO_2$, NMVOC), carbonaceous aerosols (BC and OC), and $CO_2$ developed with the Community Emissions Database System (CEDS). We improve upon existing inventories with a more consistent and reproducible methodology applied to all emissions species, updated emission factors, and recent estimates through 2014. The data system relies on existing energy consumption data sets and regional and country-specific inventories to produce trends over recent decades. All emissions species are consistently estimated using the same activity data over all time periods. Emissions are provided on an annual basis at the level of country and sector and gridded with monthly seasonality. These estimates are comparable to, but generally slightly higher than, existing global inventories. Emissions over the most recent years are more uncertain, particularly in low- and middle-income regions where country-specific emission inventories are less available. Future work will involve refining and updating these emission estimates, estimating emissions uncertainty, and publication of the system as open source software.

## 1  Introduction

Anthropogenic emissions of reactive gases, aerosols, and aerosol precursor compounds have substantially changed atmospheric composition and associated fluxes to land and ocean surfaces. As a result, increased particulate and tropospheric ozone concentrations since pre-industrial times have altered radiative balances of the atmosphere, increased human mortality and morbidity, and impacted terrestrial and aquatic ecosystems. Central to studying these effects are historical trends of emissions. Historical emissions data and consistent emissions time series are especially important for Earth Systems Models (ESMs) and atmospheric chemistry and transport models, which use emissions time series as key model inputs; Integrated Assessment Models (IAMs), which use recent emissions data as a starting point for future emissions scenarios; and to inform management decisions.

Despite their wide use in research and policy communities, there are a number of limitations to current inventory data sets. Emissions data from country and regional specific inventories vary in methodology, level of detail, sectoral coverage, and consistency over time and space. Existing global inventories do not always provide comprehensive documentation for assumptions and methods and few contain uncertainty estimates.

Several global emissions inventories have been used in global research and modeling. The Emissions Database for Global Atmospheric Research (EDGAR) is another widely used historical global emissions data set. It provides an independent estimate of historical greenhouse gas (GHG) and pollutant emissions by country, sector, and spatial grid (0.1 x 0.1 degree) from 1970 – 2010 (Crippa et al., 2016; EC-JRC/PBL, 2016), with GHG emission estimates for more recent years. The most recent set of modeling exercises by the Task Force on Hemispheric Transport of Air Pollutants (HTAP) uses a gridded emissions data set, HTAP v2 (Janssens-Maenhout et al., 2015), that merged EDGAR with regional and country-level gridded emissions data for 2008 and 2010. The GAINS (Greenhouse gas - Air pollution Interactions and Synergies) model (Amann et al., 2011) has been used to produce regional and global emission estimates for several recent years (1990- 2010; in five year intervals) together with projections to 2020 and beyond (Amann et al., 2013; Cofala et al., 2007; Klimont et al., 2009). These have been developed with substantial consultation with national experts, especially for Europe and Asia (Amann et al., 2008, 2015; Purohit et al., 2010; Sharma et al., 2015; Wang et al., 2014; Zhang et al., 2007; Zhao et al., 2013a). The newly developed ECLIPSE emission sets include several extensions and updates in the GAINS model and are also available in a gridded form (Klimont et al., 2016) and have been used in a number of recent modeling exercises (Eckhardt et al., 2015; IEA, 2016b; Rao et al., 2016; Stohl et al., 2015).

Lamarque et al. (2010) developed a historical data set for the Coupled Model Intercomparison Project Phase 5 (CMIP5), which includes global, gridded estimates of anthropogenic and open burning emissions from 1850 – 2000 at 10 year intervals. This data is also used as the historical starting point for the Representative Concentration Pathways (RCP) scenarios (van Vuuren et al., 2011) and in some research communities is referred to as the RCP historical data. In this article it is referred to as the CMIP5 data set. It was a compilation of "best available estimates" from many sources including EDGAR-HYDE (van Aardenne et al., 2001) which provides global anthropogenic emissions of carbon dioxide ($CO_2$), methane ($CH_4$), nitrous oxide ($N_2O$), nitrogen oxides ($NO_X$), non-methane volatile organic compounds (NMVOC), sulfur dioxide ($SO_2$) and ammonia ($NH_3$) from 1890 to 1990 every 10 years at 1 x 1 degree grids; RETRO (Schultz and Sebastian, 2007) which estimated global emissions from 1960 to 2000; and emissions reported

by, largely, Organization for Economic Co-operation and Development (OECD) countries over recent years. While this data set was an improvement upon the country and regional specific inventories mentioned above, it lacks uncertainty estimates and reproducibility, has limited temporal resolution (10 year estimates to 2000), and does not have consistent methods across emission species. There are many existing inventories of various scope, coverage, and quality; however, no existing data set meets all the growing needs of the modeling community.

This paper describes the general methodology and results for an updated global historical emissions data set that has been designed to meet the needs of the global atmospheric modeling community and other researchers for consistent long-term emission trends. The methodology was designed to produce annual estimates, be similar to country-level inventories where available, be complete and plausible, and use a consistent methodology over time with the same underlying driver data (e.g., fuel consumption). The data set described here provides a sectoral and gridded historical inventory of climate-relevant anthropogenic GHGs, reactive gases, and aerosols for use in the Coupled Model Intercomparison Project Phase 6 (CMIP6). It does not include agricultural waste burning, which is included in van Marle et al. (van Marle et al., 2017). Gridded data were first released summer 2016 through the Earth System Grid Federation (ESGF) system including $SO_2$, $NO_X$, $NH_3$, carbon monoxide (CO), black carbon (BC), organic carbon (OC), and NMVOC, with a new release in May 2017 that corrected mistakes in the gridded data (links and details in Appendix A1 and A2). The May 2017 release also included $CO_2$ emissions (annual from 1750 - 2014) and $CH_4$ emissions (annual from 1970 – 2014 and a separate decadal historical extension from 1850 – 1970, also detailed in Appendix A2). This data set was created using the Community Emissions Database System (CEDS), which is being prepared for release as open-source software. Updated information on the system can be found at http://www.globalchange.umd.edu/ceds/.

An overview of the methodology and data sources are provided in Sect. 2 while further details on the methodology and data sources are included in the Supplementary Information (SI), outlined in Sect. 2.7. Section 3 compares this data set to existing inventories and Sect. 4 details future work involving this data set and system.

## 2 Data and methodology

### 2.1 Methodological overview

CEDS uses existing emissions inventories, emissions factors, and activity/driver data to estimate annual country, sector, and fuel specific emissions over time in several major phases (data system schematic shown in Figure 1):

1) data are collected and processed into a consistent format and timescale (detailed in Sect. 2.2 and throughout paper),

2) default emissions from 1960/1971 (1960 for most OECD countries and 1971 for all others) to 2014 are estimated using driver and emission factor data (Emissions = Driver × Emission Factors) (Sect. 2.2),

3) default estimates are scaled to match existing emissions inventories where available, complete, and plausible (Sect. 2.4),

4) scaled emissions estimates are extended back to 1750 (Sect. 2.5) to produce final aggregate emissions by country, fuel and sector,

5) emissions are checked and summarized to produce data for release and analysis and

6) gridded emissions with monthly seasonality and VOC speciation are produced from aggregate estimates using spatial proxy data (Sect. 2.6).

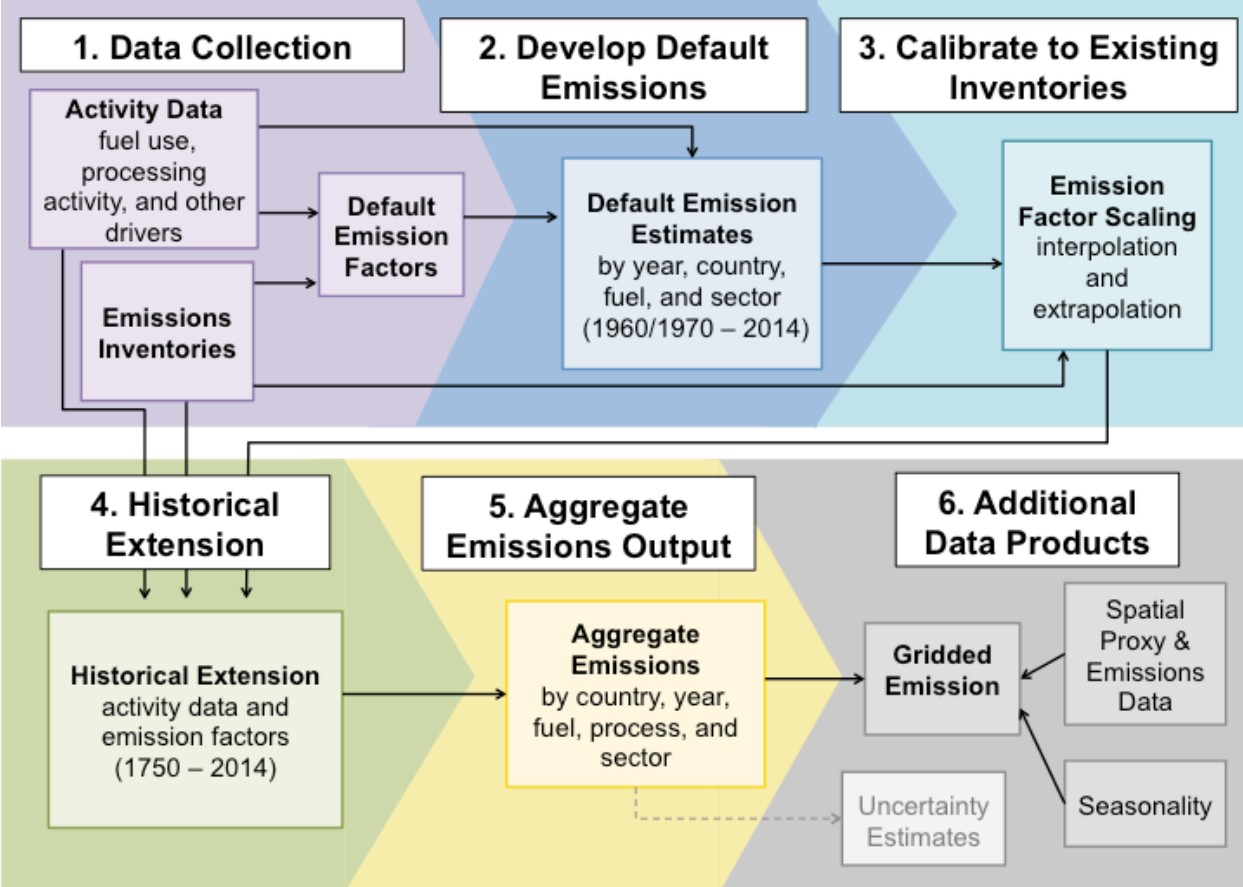

**Figure 1: System Summary. Key steps in calculation are: 1) Collect and process activity, emissions factors, and emissions data 2) Develop default emissions estimates 3) Calibrate default estimates to existing inventories 4) Extend present day emission to historical time periods 5) Summarize emissions outputs 6) Produce data products including gridded emissions and uncertainty estimates.**

Rather than producing independent estimates, this methodology relies on matching default
estimates to reliable, existing emissions inventories (emission scaling) and extending those values to historical years (historical extension) to produce a consistent historical time series. While previous work (Lamarque et al., 2010) combined different data sets then smoothed over discontinuities, CEDS produces historical trends by extending the individual components (driver data and emissions factors) separately to estimate emission trends. This method captures trends
in fuel use, technology, and emissions controls over time. Estimating emissions from drivers and emission factor components also allows the system to estimate emissions in recent years, using extrapolated emission factors and quickly released fuel use data, where detailed energy statistics and emission inventories are not yet available.

CEDS estimates emissions for 221 regions (and a global region for international shipping and aircraft), 8 fuels, and 55 working sectors, summarized in Table 1. "Regions" refers to countries, regions, territories, or islands and are listed, along with mapping to summary regions and ISO codes in the supplemental files; they will henceforth be referred to as "countries". CEDS working sectors (sectors 1A1-1A5) for combustion emissions follow the International Energy Agency (IEA) energy statistics sector definitions (Table A1). The IEA energy statistics are annually updated and the most comprehensive global energy statistics available, so this choice allows for maximal use of this data. Non-combustion emissions sectors (sectors 1A1bc and 1B-7) are drawn from EDGAR and generally follow EDGAR definitions (Table A2). Sector names were derived from Intergovernmental Panel on Climate Change (IPCC) reporting categories under the 1996 guidelines and Nomenclature for Reporting (NFR) 14 (Economic Commission for Europe, 2014) together with a short descriptive name[a].   Note that CEDS data do not include open burning, e.g. forest and grassland fires, and agricultural waste burning on fields, which was developed by van Marle et al (2017). Tables providing more detailed information on these mappings, which define the CEDS sectors and fuels, are provided in Sect. A3. We note that, while agriculture sectors include a large variety of activities, in practice in the current CEDS system these sectors largely represent $NH_3$ and $NO_X$ emissions from fertilizer application (under 3D_Soil-emissions) and manure management, due to the focus in the current CEDS system on air-pollutant emissions.

In order to produce timely emissions estimates for CMIP6, several CEDS emission sectors in this version of the system aggregate somewhat disparate processes to reduce the need for the development of detailed driver and emission factor information. For example, process emissions from the production of iron and steel, aluminum, and other non-ferrous metals are grouped together as an aggregate as 2C_Metal-production sector. Similarly, emissions from a variety of processes are reported in 2B_Chemical-industry. Also, the 1A1bc_Other-tranformation sector includes emissions from combustion related activities in energy transformation processes including coal and coke production, charcoal production and petroleum refining, but are combined in one working sector (see Sec 2.3.2). Greater disaggregation for these sectors would improve these estimates, but will require additional effort, described in Sect.5 Limitations and Future work.

The core outputs of the CEDS system are country-level emissions aggregated to the CEDS sector level. Emissions by fuel and by detailed CEDS sector are also documented within the system for analysis, although these are not released due to data confidentiality issues. Emissions are further aggregated and processed to provide gridded emissions data with monthly seasonality, detailed in Sect. 2.6.

We note that the CEDS system does not reduce the need for more detailed inventory estimates. For example, CEDS does not include a representation of vehicle fleet turnover and emission control degradation (*e.g.* the effectiveness of catalytic converters over time) or multiple fuel combustion technologies that are included in more detailed inventories.  The purpose of this system, as described further below, is to build on a combination of global emission estimation

---

[a] Sector names were derived NFR14 nomenclature via a mapping table provided by CEIP, available from: http://www.ceip.at/ms/ceip_home1/ceip_home/reporting_instructions/

frameworks such as GAINS and EDGAR, combined with country-level inventories, to produce
reproducible, consistent emissions trends over time, space, and emissions species.

**Table 1 CEDS working sectors and fuels (CEDS v2016-07-26)**

| **CEDS Working Sectors** | | |
|---|---|---|
| **Energy Production** | 1A2g_Ind-Comb-other | **RCO** |
| 1A1a_Electricity-public | 2A1_Cement-production | 1A4a_Commercial-institutional |
| 1A1a_Electricity-autoproducer | 2A2_Lime-production | 1A4b_Residential |
| 1A1a_Heat-production | 2Ax_Other-minerals | 1A4c_Agriculture-forestry-fishing |
| 1A1bc_Other-transformation | 2B_Chemical-industry | 1A5_Other-unspecified |
| 1B1_Fugitive-solid-fuels | 2C_Metal-production | **Agriculture** |
| 1B2_Fugitive-petr-and-gas | 2D_Other-product-use | 3B_Manure-management |
| 1B2d_Fugitive-other-energy | 2D_Paint-application | 3D_Soil-emissions |
| 7A_Fossil-fuel-fires | 2D_Chemical-products- | 3I_Agriculture-other |
| **Industry** | manufacture-processing | 3D_Rice-Cultivation |
| 1A2a_Ind-Comb-Iron-steel | 2H_Pulp-and-paper-food- | 3E_Enteric-fermentation |
| 1A2b_Ind-Comb-Non-ferrous-metals | beverage-wood | **Waste** |
| 1A2c_Ind-Comb-Chemicals | 2D_Degreasing-Cleaning | 5A_Solid-waste-disposal |
| 1A2d_Ind-Comb-Pulp-paper | **Transportation** | 5E_Other-waste-handling |
| 1A2e_Ind-Comb-Food-tobacco | 1A3ai_International-aviation | 5C_Waste-combustion |
| 1A2f_Ind-Comb-Non-metalic-minerals | 1A3aii_Domestic-aviation | 5D_Wastewater-handling |
| 1A2g_Ind-Comb-Construction | 1A3b_Road | |
| 1A2g_Ind-Comb-transpequip | 1A3c_Rail | 6A_Other-in-total |
| 1A2g_Ind-Comb-machinery | 1A3di_International-shipping | 6B_Other-not-in-total |
| 1A2g_Ind-Comb-mining-quarying | 1A3di_Oil_tanker_loading | |
| 1A2g_Ind-Comb-wood-products | 1A3dii_Domestic-navigation | |
| 1A2g_Ind-Comb-textile-leather | 1A3eii_Other-transp | |
| **CEDS Fuels** | | |
| Hard Coal | Light Oil | Natural Gas |
| Brown Coal | Diesel Oil | Biomass |
| Coal Coke | Heavy Oil | |

## 2.2   Activity data

Trends of energy consumption and other driver (activity) data are key inputs for estimating
emissions. When choosing data to use in this system, priority was given to consistent trends over
time rather than detailed data that might only be available for a limited set of countries or time-
span.

### 2.2.1   Energy data

Energy consumption data are used as drivers for emissions from fuel combustion. Core energy
data for 1960 - 2013 are the International Energy Agency (IEA) energy statistics, which provides
energy production and consumption estimates by detailed country, fuel, and sector from 1960 –
2013 for most OECD countries and 1971 – 2013 for non-OECD countries (IEA, 2015). While
most data sources used in CEDS are open source, CEDS currently requires purchase of this
proprietary data set. IEA data are provided at finer fuel and sector level so data are often
aggregated to CEDS sectors and fuels. Mapping of IEA products to CEDS fuels is detailed in
Sect. A4. Data for a number of small countries are provided by IEA only at an aggregate level,
such as "Other Africa" and "Other Asia", are disaggregated to CEDS countries using historical
$CO_2$ emissions data from the Carbon Dioxide Information Analysis Center (CDIAC) (Andres et
al., 2012; Boden et al., 1995). Sectoral splits for Former Soviet Union (FSU) countries are
smoothed over time to account for changes in reporting methodologies during the transition to
independent countries (see SI).

IEA energy statistics were extended to 2014 using BP Statistical Review of World Energy (BP, 2015), which is freely available online and provides annual updates of country energy totals by aggregate fuel (oil, gas and coal). BP trends for aggregate fuel consumption from 2013 to 2014 were applied to all CEDS sectors in the corresponding CEDS fuel estimates to extrapolate to 2014 energy estimates by sector and fuel from 2012 IEA values.

In a few cases, IEA energy data were adjusted to either smooth over discontinuities or to better match newer information. For international shipping, where a number of studies have concluded that IEA reported consumption is incomplete (Corbett et al., 1999; Endresen et al., 2007; Eyring et al., 2010), we have added additional fuel consumption so that total consumption matches bottom-up estimates from International Maritime Organization (IMO) (2014). For China, fuel consumption appears to be underestimated in national statistics (Guan et al., 2012; Liu et al., 2015b), so coal and petroleum consumption were adjusted to match the sum of provincial estimates as used in the MEIC inventory (Multi-resolution Emissions Inventory for China) (Li et al., 2017) used to calibrate CEDS emission estimates. Several other changes were made, such as what appears to be spurious brown coal consumption over 1971-1984 in the IEA Other Asia region and a spike in agricultural diesel consumption in Canada in 1984. All such changes are documented in CEDS source code, input files, and supplementary information provided with this article.

Residential biomass was estimated by merging IEA energy statistics and Fernandes et al. (2007) to produce residential biomass estimates by country and fuel type over 1850 - 2013. Residential biomass data were reconstructed with the assumption that sudden drops in biomass consumption going back in time are due to data gaps, rather than sudden energy consumption changes. Both IEA and Fernades et al. values were reconstructed to maintain smooth per capita (based on rural population) residential biomass use over time.

Detail on methods and assumption for energy consumption estimates are available in the Data and Assumption Supplement (SI-Text) Sect. 3.

### 2.2.2 Population and other data

Consistent historical time trends are prioritized for activity driver data. For non-combustion sectors population is generally used as an activity driver. United Nations (UN) Population data (UN, 2014, 2015) is used for 1950 – 2014, supplemented from 1960 – 2014 with World Bank population statistics (The World Bank, 2016). This series was merged with HYDE historical population data (Klein Goldewijk et al., 2010). More detail is available in SI-Text Sect. 2.1.

In this data version, population is used as the non-combustion emissions driver for all but three sectors. 5C_Waste-combustion, which includes industrial, municipal, and open waste burning, is driven by pulp and paper consumption, derived from Food and Agriculture Organization of the UN (FAO) Forestry Statistics (FAOSTAT, 2015). FAO statistics converted to per capita values were smoothed and linearly extrapolated backward in time. 1B2_Fugitive-petr-and-gas, which are fugitive and flaring emissions from production of liquid and gaseous fuels together with oil refining, is driven by a composite variable that combines domestic oil and gas production with refinery inputs, derived from IEA Energy Statistics. This same driver is also used for 1B2d_Fugitive-other-energy. More detail is available in SI-Text Sect. 2.5. While non-combustion emissions use population as an "activity driver" in calculations, emissions trends are generally determined by a combination of EDGAR and country level inventories. Final

emissions estimates, therefore, reflect recent emissions inventories where these are available, rather than population trends.

## 2.3    Default estimates

Significant effort is devoted to creating reliable default emissions estimates, including abatement measures, to serve as a starting point for scaling to match country-level inventories (Sect. 2.4)
and historical extension back to 1750 (Sect. 2.5). While most default estimates do not explicitly appear in the final data set as they are altered to match inventories (Sect. 2.4), some are not altered because inventories are not available for all regions, sectors, and species. The method for calculating default emission factors varies by sectors and regions depending on available data.

Default emissions estimates (box 2 in Figure 1), are calculated using 3 types of data (box 1 in
Figure 1): activity data (usually energy consumption or population), emissions inventories, and emissions factors, according to Eq. (1).

$$\mathbf{E}_{em}^{c,s,f,t} = \mathbf{A}^{c,s,f,t} \times \mathbf{EF}_{em}^{c,s,f,t}$$

(1)

Where E is total emissions, A is activity or driver, EF is emissions factor, em is emission species, c is country, s is sector, f is fuel (where applicable), and t is year.

In general, default emissions for fuel combustion (sectors 1A in Table 1) are estimated from emission factors and activity drivers (energy consumption), while estimates of non-combustion
emissions (sectors 1B – 7A and 1A1bc) are taken from a relevant inventory and the "implied emissions factor" is inferred from total emissions and activity drivers.

### 2.3.1    Default fuel combustion emissions

Combustion sector emissions are estimated from energy consumption estimates (Sect. 2.2), and emissions factors according to Eq. (1). Default emission factors for the combustion of fuels are
derived from existing global data sets that detail emissions and energy consumption by sector and fuel, using Eq. (2):

$$EF_{em}^{c,s,f,t} = \frac{E_{em}^{c,s,f,t}}{A^{c,s,f,t}}$$

(2)

Where EF is default emission factor, E is total emissions as reported by other inventories, A is
activity data, measured in energy consumption as reported by inventories, em is emission species, c is country, s is sector, f is fuel (where applicable), and t is year.

The main data sets used to derive emission factors are shown in Table 2. Default emission factors for $NO_X$, NMVOC, CO, and $CH_4$ are estimated from the global implementation of the
GAINS model as released for the Energy Modeling Forum 30 project (https://emf.stanford.edu/projects/emf-30-short-lived-climate-forcers-air-quality) (Klimont et al.,

2017a, 2017; Stohl et al., 2015). BC and OC emission factors from 1850 – 2000 are estimated from the latest version of the Speciated Pollutant Emission Wizard (SPEW) (Bond et al., 2007).

Emission factors for $CO_2$ emissions for coal and natural gas combustion are taken from CDIAC
(Andres et al., 2012; Boden et al., 1995), with an additional coal mass balance check, as further described in SI-Text Sect. 5.4. For coal in China a lower oxidation fraction of 0.96 was assumed, see discussion in the SI-Text (Liu et al., 2015b). Because CEDS models liquid fuel emissions by fuel grade (light, medium, heavy), we use fuel-specific emission factors for liquid fuels also described in SI-Text Sect. 5.4.

Emission data are aggregated by sector and fuel to match CEDS sectors, while calculated emission factors from more aggregate data sets are applied to multiple CEDS sectors, fuels, or countries. When incomplete time series are available, emission factors are generally assumed constant back to 1970 linearly interpolated between data points, and extended forward to 2014 using trends from GAINS to produce a complete times series of default emission factors. Many
of these interpolated and extended values are later scaled to match county inventories (Sect. 2.4).

Most of the default emission factors are derived from sources that account for technology efficiencies and mitigation controls over time, but some are estimated directly from fuel properties (e.g., fuel sulfur content for $SO_2$ emissions). A control percentage is used to adjust the emission factor in these cases. In the data reported here the control percentage is primarily used
in $SO_2$ calculations (see SI-Text Sect. 5.1) where the base emission factor is derived directly from fuel properties; however, this functionality is available when needed for other emission species. In most of these cases emissions are later scaled to match inventory data.

**Table 2 Data Sources used to estimate default emissions factors for fuel combustion and default emissions from non-**
**combustion sectors**

| Source Sector | Emission Species | Data Source |
|---|---|---|
| Fuel Combustion (1A) | $NO_X$, NMVOC, CO, $CH_4$ | GAINS energy use and emissions (Klimont et al., 2017a; Stohl et al., 2015). |
| | BC, OC | SPEW energy use and emissions (Bond et al., 2007) |
| | $SO_2$ | (Europe) GAINS sulfur content and ash retention (Amann et al., 2015; IIASA, 2014a, 2014b). Smith et al. (2011) and additional sources for other regions (SI-Text 5.1) |
| | $NH_3$ | US NEI energy use and emissions (US EPA, 2013) |
| | $CO_2$ | CDIAC (Boden et al., 2016) and additional data sources |
| Fugitive Petroleum and Gas (1B) | All | EDGAR emissions(EC-JRC/PBL, 2016), ECLIPSE V5a (Stohl et al., 2015) |
| Cement (2A1) | $CO_2$ | CDIAC (Boden et al., 2016) |
| Agriculture Sectors (3) | $CH_4$ | For Sectors 3B_Manure-management, 3B_Soil-emissions, and 3D_Rice-Cultivation: FAOSTAT (FAO, 2016) all others: EDGAR emissions (EC-JRC/PBL, 2016) |
| | Other | EDGAR emissions (EC-JRC/PBL, 2016) |

| Waste Combustion (5C) | All | (Akagi et al., 2011; Andreae and Merlet, 2001; Wiedinmyer et al., 2014) (SI-Text Sect. 6.3) |
|---|---|---|
| Waste Water Treatment (5D) | $NH_3$ | CEDS estimate of $NH_3$ from human waste (SI-Text Sect. 6.4) |
| Other Non-Combustion | $SO_2$ | EDGAR, (Smith et al., 2011) & other sources (SI-Text Sect. 6.5) |
| (2A – 7A) | Other | EDGAR emissions (EC-JRC/PBL, 2016) |

### 2.3.2  Default non-combustion emissions

Default non-combustion emissions, are generally taken from existing emissions inventories, primarily EDGAR (EC-JRC/PBL, 2016) and some additional sources for specific sectors detailed in Table 2. Default emissions from sectors not specifically called out in Table 2 or the text below are taken from EDGAR (EC-JRC/PBL, 2016). Other data sources and detailed methods are explained in the SI-Text Sect. 6. For detailed sector definitions refer to Sect. A3.

When complete trends of emissions estimates are not available, they are extended in a similar manner as combustion emissions: emission factors are inferred using Eq. (2) and (with few exceptions) using population as an activity driver; emission factors (e.g. per-capita emissions) are linearly interpolated between data points and extended forward and back to 1970 and 2014 to create a complete trend of default emission factors; and default emissions estimates are calculated using Eq. (1).

For this data set, all non-combustion sectors (except for 5C_Waste-combustion) use population as the activity driver since this provides a continuous historical time series to be used where interpolations were needed.  In practice, since EDGAR is generally used for default non-combustion data source, we are relying on EDGAR trends by country to extend emissions data beyond years where additional inventory information does not exist (with exceptions as noted in Table 2). Sector uses pulp and paper consumption, detailed in Sect. 2.2; while the waste combustion sector, which incorporates solid waste disposal (incineration) and residential waste combustion, and is the product of combustion, in this system it is methodologically treated as a non-combustion sector.

We note that, while emissions from sector 1A1bc_Other_tranformation are also due to fuel combustion, due to the complexity of the processes included, this sector is treated as a non-combustion sector in CEDS in terms of methodology. This means that fuel is not used as an activity driver and that default emissions for this sector are taken from SPEW for BC and OC and EDGAR for other emissions. The major emission processes in this sector include coal coke production, oil refining, and charcoal production. A mass balance calculation for $SO_2$ and $CO_2$ focusing on coal transformation was also conducted to assure that these specific emissions were not underestimated, particularly for periods up to the mid 20[th] century (SI-Text Sect. 5.4, 6.5.2, and 8.3.2).

During the process of emissions scaling we found that default emissions were sometimes 1-2 orders of magnitude different from emissions reported in national inventories. This is not surprising, since non-combustion emissions can be highly dependent on local conditions, technology performance, and there are also often issues of incompleteness of inventories. In these cases, we implemented a process whereby default non-combustion emissions were taken


directly from national inventories, and gap-filled and trended over time using EDGAR estimates. These were largely fugitive and flaring emissions (1B) for $SO_2$; soil(3D), manure(3B), and waste water(5D) emissions for $NH_3$; and non-combustion emissions for NMVOCs, typically associated with solvent use.

## 2.4 Scaling emissions

CEDS uses a "mosaic" strategy to scale default emissions estimates to authoritative country-level inventories when available. The goal of the scaling process is to match CEDS emissions estimates to comparable inventories while retaining the fuel and sector detail of the CEDS estimates. The scaling process modifies CEDS default emissions and emission factors, but activity estimates remain the same.

A set of scaling sectors is defined for each inventory so that CEDS and inventory sectors overlap. These sectors are chosen to be broad, even when more inventory detail is available, because it is often unclear if sector definitions and boundaries are comparable between data sets. For example, many inventories do not consistently break out industry auto-producer electricity from other industrial combustion, so they are combined together for scaling. Additionally, underlying driver data in inventories and CEDS may not match. Scaling detailed sectors that were calculated using different energy consumption estimates would yield unrealistic scaled emission factors at a detailed sector level. One example is off-road emissions; while often estimated in country inventories, energy consumption data at this level is not consistently available from the IEA energy statistics, so these emissions are combined into broader sector groupings, depending on the sector categories available in a specific inventory.

The first step in this process is to aggregate CEDS emissions and inventory emissions to common scaling sectors, then scaling factors are calculated with Eq. (3). Scaling factors represent the ratio between CEDS default estimates and scaling inventory estimates by scaling sector and provide a means for matching CEDS default estimates to scaling inventories.

$$SF_{em}^{c,ss,t} = \frac{Inv_{em}^{c,ss,t}}{CEDS_{em}^{c,ss,t}}$$

**(3)**

Where SF is scaling factor, Inv is the inventory emissions estimate, CEDS is the CEDS emissions estimate, em is emission species, c is country, ss is aggregate scaling sector (unique to inventory), and t is year.

For each inventory, scaling factors are calculated for years when inventory data are available. Calculated scaling factors are limited to values between 1/100 and 100. Scaling factors outside this range may result from discontinuities or misreporting in inventory data; imperfect scaling maps between CEDS sectors, inventory sectors, and scaling sectors; or default CEDS emissions estimates that are drastically different than reported inventories. Many of these cases were resolved by using the detailed inventory data as default emissions data, as noted above in Sect 2.3.2. Where inventory data are not available over a portion of the specified scaling timeframe, remaining scaling factors are extended, interpolated between to provide a continuous trend. Scaling factors are applied to corresponding CEDS default emissions estimates and default emission factors to produce a set of scaled emissions components (total emissions and emission

factors, together with activity drivers, which are not changed), which are used in the historical extension (Sect. 2.5). Using scaling factors retains the sector and fuel level detail of CEDS
default emissions estimates, while matching total values to authoritative emissions inventories.

We use a sequential methodology in which CEDS values are generally first scaled to EDGAR (EC-JRC/PBL, 2016) for most emission species, then national inventories, where available. Final CEDS results, over the period these inventories were available, match the last inventory scaled. $SO_2$, $CH_4$, BC, and OC are not scaled to EDGAR values. For all pollutant species other than BC
and OC, estimates are then scaled to match country-level emissions estimates. These are available for most of Europe through European Monitoring and Evaluation Programme (EMEP) for European countries post 1980 (EMEP, 2016); the United Nations Framework Convention on Climate Change (UNFCCC) GHG data for Belarus, Greece and New Zealand (UNFCCC, 2015) post 1990; an updated version of Regional Emissions Inventory in Asia (REAS) for Japan
(Kurokawa et al., 2013a); Multi-resolution Emissions Inventory for China (MEIC) for China (Li et al., 2017); and others detailed in Table 3. BC and OC emissions estimates are entirely from default estimates calculated using predominantly SPEW data. While BC inventory estimates were available in a few cases, OC estimates were less available, so we have retained the consistent BC and OC estimates from SPEW for all countries. $CH_4$ emissions estimates are
scaled to match to the following inventories: EDGAR 4.2 (EC-JRC/PBL, 2012), UNFCCC submissions UNFCCC, 2015) for most "Annex I" countries, and the US GHG inventory (US EPA, 2012b) for the United States.

The scaling process was designed to allow for exceptions when there are known discontinuities in inventory data or when the default scaling options resulted in large discontinuities. For
example, Former Soviet Union countries were only scaled to match EDGAR and other inventories after 1992 (where energy data becomes more consistent). Romania, for example, was only scaled to match EDGAR in 1992, 2000, and 2010 to avoid discontinuities. For the most part, these exceptions occur for countries with rather limited penetration of control measures or only low efficiency controls. Regions with more stringent emission standards requiring extensive
application of high efficiency controls have typically higher quality national inventories, e.g., European Union, North America, and parts of Asia.

Description of the exceptions and assumptions for all scaling inventories, as well as a detailed example of the scaling process is available in SI-Text Sect. 7. Additionally, figures showing stacked area graphs of global emission, by final scaling inventory (or default estimate) are shown
in Supplemental Figures and Tables Sect. J. These show the percentage of final global emissions estimates that are scaled to various inventories.

**Table 3 Data Sources for Inventory Scaling. All countries scaled to EDGAR, then individual estimates.**

| Region/ Country | Species | Years | Data Source |
|---|---|---|---|
| All, where available | $NO_X$, NMVOC, CO, $NH_3$ | 1970 - 2008 | EDGAR 4.3 (EC-JRC/PBL, 2016) |
| | $CH_4$ | 1970 - 2008 | EDGAR 4.2 (EC-JRC/PBL, 2012) |

| | | | |
|---|---|---|---|
| Europe | SO$_2$, NO$_X$, NMVOC, CO, NH$_3$ | 1980 - 2012 | (EMEP, 2016) |
| Greece, New Zealand, Belarus | SO$_2$, NO$_X$, NMVOC, CO, CO2 | 1990 - 2012 | (UNFCCC, 2015) |
| Other Asia | SO$_2$, NO$_X$, NMVOC, CO, CH4 | 2000 - 2008 | REAS 2.1 (Kurokawa et al., 2013a) |
| Argentina | SO$_2$, NO$_X$, NMVOC, CO | 1990 - 1999, 2001 - 2009, 2011 | (Argentina UNFCCC Submission, 2016) |
| Australia | SO$_2$, NO$_X$, NMVOC, CO | 2000, 2006, 2012 | (Australian Department of the Environment, 2016) |
| China | SO$_2$, NO$_X$, NMVOC, CO, NH$_3$ | 2008, 2010, 2012 | MEIC (Li et al., 2017) |
| Canada | SO$_2$, NO$_X$, NMVOC, CO | 1985 - 2011 | (Environment and Climate Change Canada, 2016; Environment Canada, 2013) |
| Japan | SO$_2$, NO$_X$, NMVOC, CO, NH$_3$ | 1960 - 2010 | Preliminary update of Kurokawa et al., (2013b) |
| South Korea | SO$_2$, NO$_X$, NMVOC, CO | 1999 - 2012 | (South Korea National Institute of Environmental Research, 2016) |
| Taiwan | SO$_2$, NO$_X$, NMVOC, CO | 2003, 2006, 2010 | (TEPA, 2016) |
| USA | SO$_2$, NO$_X$, NMVOC, CO | 1970,1975,1980,1985, 1990 - 2014 | EPA Trends (US EPA, 2016b) |
| | NH$_3$ | 1990 - 2014 | |
| | CO$_2$ | 1990 - 2014 | US EPA, 2016a) |
| | CH$_4$ | 1990 - 2014 | US GHG Inventory (US EPA, 2012b) |

The scaling process operates on sectors where emissions are present in both the CEDS default data and the scaling inventories listed in Table 3. If the scaling inventory does not contain information for a particular sector then the default data are used. This means that some gaps in the scaling inventories are automatically filled by this procedure and, as a result, the CEDS emission totals can be larger than those in the scaling inventory. For example, waste burning and
fossil fuel fires are not included in some of the inventories, while these sectors are included in CEDS. In a few cases, specific additional data were added where gaps were known to be present. For example, the CEDS totals for China are slightly larger than the MEIC totals due to both the inclusion of waste burning, but also the addition of SO$_2$ emissions from metal smelting, which are not included in MEIC. Where necessary discontinuities in inventory estimates were
eliminated. For the USA, for example, discontinuities were present in the original EPA trends data due to methodological changes, particularly for transportation NO$_x$ and agricultural NH$_3$.

## 2.5   Pre-1970 emissions extension

Historical emissions and energy data before 1970 generally do not have the same detail as more modern data. In general, we extend activity and emission factors back in time separately, with time and sector specific options to capture changes in technologies, fuel mixes, and activity. This allows for consistent methods across time and sectors, rather than piecing together different sources and smoothing over discontinuities, which was done in previous work (Lamarque et al., 2010). For most emission species and sectors the assumed historical trend in activity data has a large impact on emission trends. Activity for many sectors and fuels, such as fossil liquid and gas fuels, are small or zero by 1900. Some cases where emission factors are known to have changed over time have also been incorporated.

### 2.5.1 Pre-1970 activity drivers

IEA Energy Statistics, which are the foundation for energy estimates in this data set, go back to 1960 at the earliest. Fossil fuels are extended using CDIAC emissions, SPEW energy data, and assumptions about fuel type and sector splits in 1750, 1850, and 1900, detailed in the SI-Text Sect. 8.1. First total fuel use for three aggregate fossil fuel types, coal, oil, and gas, are estimated over 1750 - 1960/1970 for each country using historical national $CO_2$ estimates from the Carbon Dioxide Information Analysis Center (CDIAC) (Andres et al., 1999; Boden et al., 2016).

For coal only, these extended trends were matched with SPEW estimates of total coal use, which are a composite of UN data (UN, 2016) and Andres et al., (1999). This resulted in a more accurate extension for a number of key countries. SPEW estimates at every 5 years were interpolated to annual values using CDIAC CO2 time series, resulting in an annual time series. For coal and petroleum, aggregate fuel use was disaggregated into specific fuel types (e.g., brown coal, hard coal and coal coke; light, medium, and heavy oil) by smoothly transitioning between fuel splits by aggregate sector from the IEA data to SPEW fuel type splits in earlier time periods. Finally fuel use was disaggregated into sectors in a similar manner, smoothly transitioning between CEDS sectoral splits in either 1970 or 1960 to SPEW sectoral splits by 1850. A number of exogenous assumptions about fuel and sector splits over time were also needed in this process. More detail on this method can be found in supplement SI-Text Sect. 8.1.1.

While most biomass fuels are consumed in the residential sector, whose estimation was described above (Sect. 2.2.1), biomass consumed in other sectors are extended using SPEW energy data and population. 1970 CEDS estimates of biomass used in industrial sectors are merged to SPEW values by 1920. Biomass estimates from 1750 – 1850 are estimated by assuming constant per-capita values.

Activity drivers for non-combustion sectors in modern years are primarily population estimates. Most historical drivers for non-combustion sectors are also population, while some, shown in Table 4, are extended with other data. These are mostly sectors related to chemicals and solvents that are extended with $CO_2$ trends from liquid fuel use. Waste combustion is estimated by historical trends for pulp and paper consumption. The driver for sectors 1B2 and 1B2d, refinery and natural gas production, is extended using CDIAC $CO_2$ emissions for liquid and gas fuels.

**Table 4 Historical Driver Extensions for Non-Combustion Sectors**

| Non-Combustion Sector | Modern Activity Driver | Historical Extension Trend |
| --- | --- | --- |

| 1B2_Fugitive-petr-and-gas | Refinery and natural gas production | CDIAC – liquid and gas fuels $CO_2$ |
|---|---|---|
| 1B2d_Fugitive-other-energy | Refinery and natural gas production | CDIAC – liquid and gas fuels $CO_2$ |
| 2B_Chemical-industry | population | CDIAC – liquid fuels $CO_2$ |
| 2D_Degreasing-Cleaning | population | CDIAC – liquid fuels $CO_2$ |
| 2D_Paint-application | population | CDIAC – liquid fuels $CO_2$ |
| 2D3_Chemical-products-manufacture-processing | population | CDIAC – liquid fuels $CO_2$ |
| 2D3_Other-product-use | population | CDIAC – liquid fuels $CO_2$ |
| 2L_Other-process-emissions | population | CDIAC – liquid fuels $CO_2$ |
| 5C_Waste-combustion | Pulp and paper consumption | |
| 7A_Fossil-fuel-fires | population | CDIAC – cumulative solid fuels $CO_2$ |
| All Other Process Sectors | population | |

## 2.5.2  Pre-1970 emission factors

In 1850, the only fuels are coal and biomass used in residential, industrial, rail, and international shipping sectors, and many non-combustion emissions are assumed to be zero. Emission factors are extended back in time by converging to a value in a specified year (often 0 in 1850 or 1900), remaining constant, or following a trend. For some non-combustion emissions, we use an emission trend instead of an emission factor trend. Ideally, sector-specific activity drivers would extend to zero, rather than emissions factors; however, we often use population as the activity driver, because of the lack of complete, historical trends. Extending the emissions factor (e.g., the per capita value) to zero approximates the decrease to zero in the actual activity.

BC and OC emission factors for combustion sectors were extended back to 1850 by sector and fuel using the SPEW database and held constant before 1850. Combustion emission factors for $NO_X$, NMVOC, and CO in 1900 are drawn from a literature review, primarily Winijkul et al (2016). These emission factors were held constant before 1900 and linearly interpolated between 1900 and 1970. Additional data sources and details are available in the SI-Text Sect. 8.2.

Many non-combustion emissions were trended back with existing data from the literature. These include trends from SPEW (Bond et al., 2007), CDIAC (Boden et al., 2016), sector specific sources such as $SO_2$ smelting and pig iron production, and others, detailed in Table 5. Emissions factors for remaining sectors were linearly interpolated to zero in specified years based on a literature review ((Bond et al., 2007; Davidson, 2009; Holland et al., 2005; Smith et al., 2011)). Further methods and data sources are found in SI-Text Sect. 8.3.

$NH_3$ and $NO_X$ emissions from mineral and manure (3B_Manure-management and 3D_Soil-emissions) are grouped together. While CEDS total estimates should be reliable, there might be inconsistencies going back in time. We assume that the dominant trend from 1960 to 1970 is mineral fertilizer, then scaled back in time globally using Davidson et al. (2009).

**Table 5 Historical Extension Method and Data Sources for Emission Factors**

| Sector | Emission Species | Extension Method | Data Source |
|---|---|---|---|
| All Combustion Sectors | NMVOC, CO, NO$_X$ | Interpolate to value in 1900 | Detailed in SI-Text (Sect. 8.2.1) |
| All Combustion Sectors | BC, OC | EF trend | SPEW |
| 2Ax_Other-minerals, 2D_Degreasing-Cleaning, 2D_Paint-application, 2D3_Chemical-products-manufacture-processing, 2D3_Other-product-use, 2H_Pulp-and-paper-food-beverage-wood, 2L_Other-process-emissions, 5A_Solid-waste-disposal, 5C_Waste-combustion, 5E_Other-waste-handling, 7A_Fossil-fuel-fires | All | Interpolate to zero in specified year [EFs are emissions per capita values] | Detailed in SI-Text (Sect. 8.3.1) |
| 5D_Wastewater-handling, | NH$_3$ | Interpolate to value in specified year | |
| 3B_Manure-management | NH3, NO$_X$ | EF trend Emissions trend | Manure Nitrogen per capita (Holland et al., 2005) See SI-Text (Sect. 8.3.1) |
| 3D_Soil-emissions | NH3, NO$_X$ | EF trend Emissions trend | 1961-1970: Emissions trend using total nitrogen (N) fertilizer by country 1860-1960: per-capita emissions scaled by global N fertilizer (Davidson, 2009) See Supplemental Information (Sect. 8.3.1) |
| 1A1a_Electricity-public, 1A1a_Heat-production, 1A2g_Ind-Comb-other, 1A3c_Rail, 1A4a_Commercial-institutional, 1A4b_Residential | SO2 | EF trend | (Gschwandtner et al., 1986) |
| 1A1bc_Other-transformation | BC, OC | Emissions Trend | Pig iron production (SPEW, USGS, other) |
| 1A1bc_Other-transformation | others | Emissions Trend | Total fossil fuel CO2 (CDIAC) |
| 2A1_Cement-production, 2A2_Lime-production | - | Emissions Trend | CDIAC Cement CO2 |
| 2C_Metal-production | SO2 | Emissions Trend | Smith et al. (2011) Emissions |
| 2C_Metal-production | CO | Emissions Trend | Pig iron production |
| 2C_Metal-production | others | Emissions Trend | CDIAC solid fuel CO2 |

## 2.6   Gridded emissions

Final emissions are gridded to facilitate use in Earth system, climate, and atmospheric chemistry models. Gridded outputs are generated as CF-compliant NetCDF files (http://cfconventions.org/).
Aggregate emissions by country and CEDS sector are aggregated to 16 intermediate sectors (Table 6) and downscaled to a 0.5 x 0.5 degree grid. Country-aggregate emissions by intermediate gridding sector are spatially distributed using normalized spatial proxy distributions

for each country, plus global spatial proxies for shipping and aircraft, then combined into global maps. For grid cells that contain more than one country, the proxy spatial distributions are
adjusted to be proportional to area fractions of each country occupying that cell. Gridded emissions are aggregated to 9 sectors for final distribution: agriculture, energy, industrial, transportation, residential/commercial/other, solvents, waste, international shipping, and aircraft (shown in Table 6, more detail in SI-Text Sect 9.1).

Proxy data used for gridding are primarily gridded emissions from EDGAR v4.2(EC-JRC/PBL,
2012) and HYDE population (Goldewijk et al., 2011). Flaring emissions use a blend of grids from EDGAR and ECLIPSE (Klimont et al., 2017a). Road transportation uses the EDGAR 4.3 road transportation grid, which is significantly improved over previous versions (EC-JRC/PBL, 2016), but was only available for 2010, so this is used for all years. When the primary proxy for a specific country/region, sector, and year combination is not available, CEDS uses gridded
population from Gridded Population of the World (GPW) (Doxsey-Whitfield et al., 2015) and HYDE as backup proxy. Whenever available, proxy data are from annual gridded data, however proxy grids for sectors other than RCO (residential, commercial, other) and waste are held constant before 1970 and after 2008. Specific proxy data sources are detailed in Table 6. As noted above, these proxy data were used to distribute emissions spatially within each country
such that country totals match the CEDS inventory estimates. More detail on gridding can be found in the SI-Text Sect. 9.

Emissions are aggregated to 9 final gridding sectors (Table 6) and distributed over 12 months using spatially-explicit, sector-specific, monthly fractions, largely from the ECLIPSE project, except for international shipping (from EDGAR) and aircraft (from Lee et al. (2009), as used in
Larmarque et al. 2010). Emissions are then converted to flux (kg m$^{-2}$s$^{-1}$). This process is further described in the SI-Text Section 9.4.

**Table 6 Proxy Data used for Gridding**

| CEDS final gridding sector | CEDS intermediate gridding sector definition | Proxy Data Source | Years |
|---|---|---|---|
| Residential, commercial, other (RCO) | Residential, Commercial, Other (Residential and Commercial) | HYDE Population (Decadal values, interpolated annually) | 1750 - 1899 |
| | | EDGAR v4.2 (1970) blended with HYDE Population | 1900 - 1969 |
| | | EDGAR v4.2 RCORC | 1970 – 2008 |
| | Residential, Commercial, Other (Other) | HYDE Population (Decadal values, interpolated annually) | 1750 - 1899 |
| | | EDGAR v4.2 (1970) blended with HYDE Population | 1900 - 1969 |
| | | EDGAR v4.2 RCOO | 1970 – 2008 |
| Agriculture (AGR) | Agriculture | EDGAR v4.2 AGR | 1970 – 2008 |
| Energy sector (ENE) | Electricity and heat production | EDGAR v4.2 ELEC | 1970 – 2008 |
| | Fossil Fuel Fires | EDGAR v4.2 FFFI | 1970 – 2008 |
| | Fuel Production and Transformation | EDGAR v4.2 ETRN | 1970 – 2008 |

| | | | |
|---|---|---|---|
| | Oil and Gas Fugitive/Flaring | ECLIPSE FLR 1990, 2000, 2010<br>EDGAR v4.2 ETRN (1970 - 2008) | 1970 – 2010 |
| Industrial sector (IND) | Industrial Combustion | EDGAR v4.2 INDC | 1970 – 2008 |
| | Industrial process and product use | EDGAR v4.2 INPU | 1970 – 2008 |
| Transportation section (TRA) | Road Transportation | EDGAR v4.3 ROAD (2010) | 1750 – 2014 |
| | Non-road Transportation | EDGAR v4.2 NRTR | 1970 – 2008 |
| International shipping (SHP) | International Shipping | ECLIPSE + additional data (1990 – 2015) | 1990 - 2010 |
| | International Shipping (Tanker Loading) | ECLIPSE + additional data (1990 – 2015) | 1990 - 2010 |
| Solvents production and application (SLV) | Solvents production and application | EDGAR v4.2 SLV | 1970 – 2008 |
| Waste (WST) | Waste | HYDE Population, GPW v3 (modified rural population) | 1750 – 2014 |
| Aircraft (AIR) | Aircraft | CMIP5 (Lamarque et al., 2010; Lee et al., 2009) | 1850 - 2008 |
| * Spatial proxy data within each country is held constant before and after the years shown. See Supplement for further details on the gridding proxy data including definitions for the EDGAR gridding codes in this table. | | | |

**2.7   Additional methodological detail**

The above sections discuss the general approach to the methodology used in producing this data set, but there are a number of exceptions, details on additional processing and analysis, and data sources that are discussed in the Supplemental files.

**3   Results and discussion**

**3.1   Emissions trends**

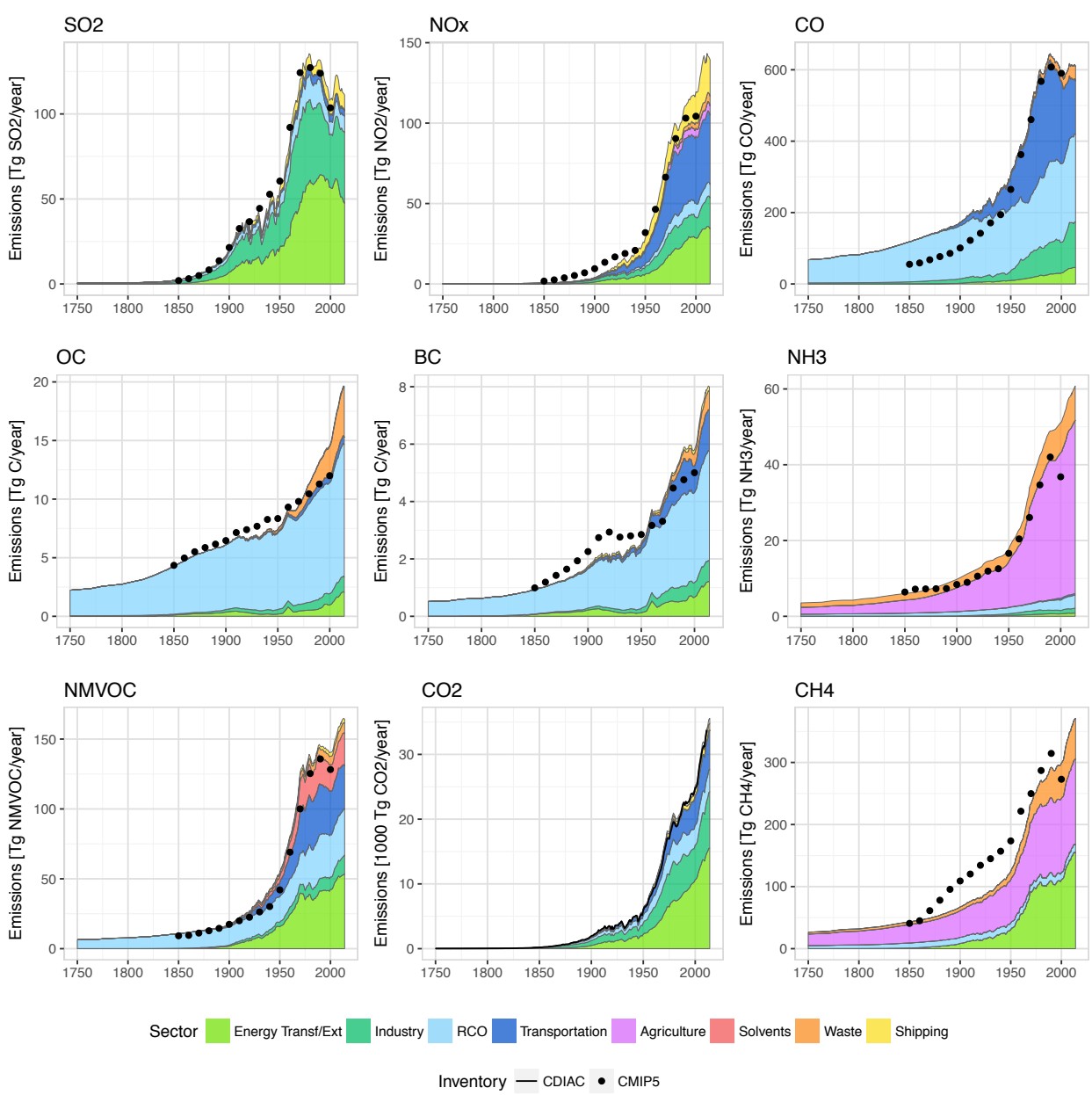

**Figure 2: CEDS emissions estimates by aggregate sector compared to Lamarque et al. (2010) (dots) and CDIAC (line) for CO₂. For a like with like comparison, these figures do not include aviation or agricultural waste burning on fields. 'RCO' stands for residential, commercial, and other.**


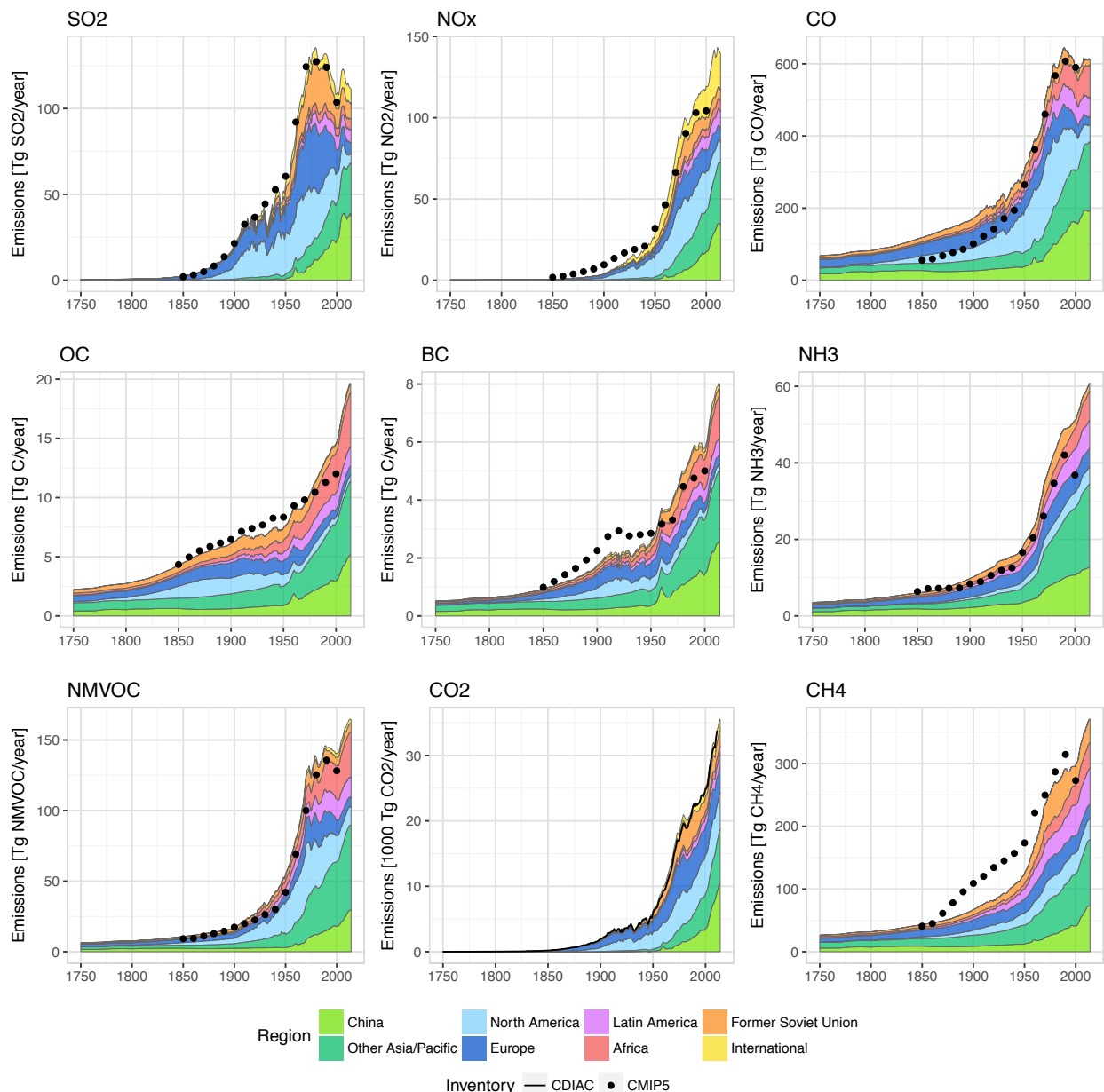

**Figure 3: Emissions estimates by region compared to Lamarque et al. (2010) (dots) and CDIAC (line) for CO₂. For a like with like comparison, these figures do not include aviation or agricultural waste burning on fields. The "International" region shows international shipping emissions.**

Figure 2 and Figure 3 show global emissions over time by aggregate sector and region, respectively, from 1750 – 2014. Definitions of aggregate sectors and regions are given in Supplemental Figure and Tables, Sect. A. The supplement Sect. B contains line graph versions of these figures, emissions by fuel, and regional versions of Figure 2 and Figure 3.

In 1850, the earliest year in which most existing data sets provide estimates, anthropogenic
emissions are dominated by residential sector cooking and heating and therefore products of incomplete combustion for BC, OC, CO, and NMVOC. In 1850, anthropogenic emissions (sectors included in this inventory), make up approximately 20 – 30% of total global emissions

(which also include grassland and forest burning, estimated by Lamarque et al. (2010)) for BC, OC, NMVOC, and CO but only 3% of global $NO_X$ emissions.

In the late 1800s through mid 20[th] century, global emissions transition to a mix of growing industrial, energy transformation and extraction (abbreviated as "Energy Trans/Ext"), and transportation emissions with a relatively steady global base of residential emissions (primarily biomass and later coal for cooking and heating). The 20[th] century brought a strong increase in emissions of pollutants associated with the industrial revolution and development of the transport

sectors ($SO_2$, $NO_x$, $CO_2$, NMVOC). BC and OC exhibit steadily growing emissions dominated by the residential sector over the century, while other sectors begin to contribute larger shares post 1950. The last few decades increasingly show, even at the global level, the impact of strong growth of Asian economies (Fig. 3). The Haber-Bosch invention (ammonia synthesis) about 100 years ago allowed fast growth in agricultural production, stimulating population growth and a

consequent explosion of $NH_3$ emissions (Erisman et al., 2008). Before 1920 global emissions for all species are less than 10% of year 2000 global values.

For several decades after 1950 global emissions grow quickly for all species. $SO_2$ continues to be dominated by industry and energy transformation and extraction sectors. In the later parts of the

century, while Europe and North American $SO_2$ emissions decline as a result of emission control policies, $SO_2$ emissions in Asia continue to grow. $NH_3$ is dominated by the agriculture sectors and NMVOCs by industry and energy transformation and extraction sectors. Transportation emissions have grown steadily and became an important contribution to $NO_X$, NMVOC, and CO emissions. Growth in CO emissions over the century is due to transportation emission globally

until the 1980s and 90s when North America and Europe introduced catalytic converters. Other regions followed more recently resulting in a declining transport contribution, however, CO emissions in Asia and Africa have continued to rise due to population-driven residential biomass burning. Similarly, while $NO_X$ from transportation sectors have decreased in recent years, total global $NO_X$ emissions have increased quickly since 2005 due to industry and energy sectors in

all parts of Asia. BC and OC increases since 1950 have been dominated by residential emissions from Africa and Asia but growing fleets of diesel vehicles in the last decades added to the burden of BC emissions.

BC emissions from residential biomass are shown in Figure 4 alongside rural population by

region. Other Asia, Africa, and China dominate residential biomass BC emissions, which are regions with the largest rural populations. While residential biomass in most regions follow rural population trends, emissions in Latin America stay flat while its rural population has steadily increased since 1960. Emissions in China flatten more dramatically after 1990 than rural population, presumably reflecting the spread of modern energy sources as rural residential per

capita biomass use decreases in this dataset.

Of the emission species estimated, $SO_2$ is the most responsive to global events such as war and depressions. $SO_2$ emissions are primarily from non-residential fuel burning and industrial processes which vary with economic activity, where other species have a base of residential

biomass burning or agriculture and waste emissions. In this data set, these emissions remain steady within the backdrop of variable economic conditions, while events such as World Wars or the collapse of the Soviet Union can be seen most clearly in annual $SO_2$ emissions. We note that the relative constancy of residential and agricultural emissions is, to some extent, a result of a

lack of detailed time series data for the drivers of these emissions in earlier periods. Variability
for these sectors in earlier years, therefore, might be underestimated.

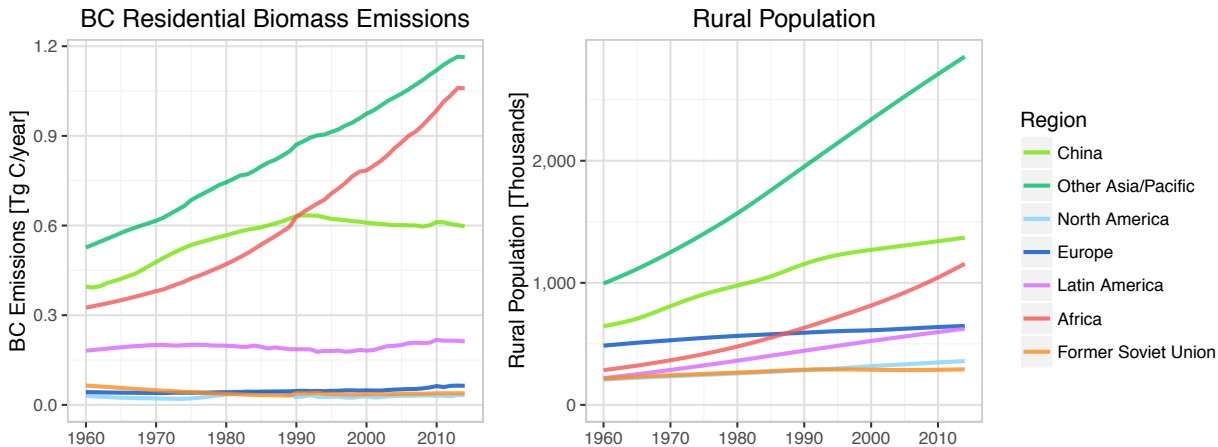

**Figure 4: (Left) BC residential biomass emissions by region and (right) rural population by region.**

### 3.2    Emissions trends in recent years (2000 - 2014)

After 2000, many species' emissions follow similar trends as the late 20[th] Century, as shown in
Figure 5, with further details in the SI-Figures Sect. C, E, and G.

BC and OC steadily grow in Africa and Other Asia from residential biomass emissions, which
are driven by continued growth of rural populations. While most BC emission growth in China is
due to energy transformation, primarily coke production, the residential, transportation, industry
and waste sectors all contribute smaller, but similar growth over 2000 – 2014 (Fig. S19). See
Sect. 3.4 for a discussion of uncertainty.

$NH_3$ continues its steady increase mostly due to agriculture in Asia and Africa. Global $CO_2$
emissions increase due to steadily rising emissions across most sectors in China and Asia and
moderately rising emissions in Africa and Latin America, while emissions in North America and
Europe flatten or decline after 2007 (largely due to the energy transformation and extraction
sectors).

Global CO emissions flatten, despite increasing CO emissions in China and Other Asia, and
Africa, which is offset by a continuing decrease of transportation CO emissions in North
America and Europe, shown in Fig. 2 and in more detail in the Supplemental Figures. CO
emissions in China increase then flatten after 2007, despite continually decreasing transportation
CO emissions, which are offset by an increase in industrial emissions (Fig. S19). Similarly, after
an increase from 2000 – 2005, global $SO_2$ emissions flatten despite increasing emissions in
China and Other Asia due to steadily decreasing emissions in Europe, North America, and the
Former Soviet Union (Figures 2 and S3). $SO_2$ emissions from energy transformation in China
have declined since 2005 with the onset of emissions controls in power plants, however
industrial emissions remained largely uncontrolled and became the dominant sector in China
(Fig. S19).

Global $NO_X$ emissions rise then flatten around 2008. The growth in industrial emissions after
2000 is offset in 2007 by the decrease in international shipping emissions, while global

emissions in other sectors stay flat. $NO_X$ emissions in North America and Europe decline due to transportation and energy transformation (Simon et al., 2015), while emissions in China and Other Asia continue to grow, also in the transportation and energy transformation. Growth of $NO_X$ emissions in Other Asia, almost completely offset reductions in $NO_X$ emissions in North America from 2000 – 2014. In China, industry continually grows since 2003, transportation

began to flatten around 2007, and the energy transformation and extraction sectors began declining in 2011 (Fig. S19) following the introduction of more stringent emission standards for power plants (Liu et al., 2016).

Globally, NMVOC emissions increase over the period, due to varying developments across the regions but in large part due to increases in energy emissions. NMVOC emissions increase in

China from solvents (Fig. S19), Other Asia from transportation (Fig. S24), and Africa from energy transformation (Fig. S18); decline in Europe and North America due to transportation and solvents (Fig. S20 and S23), and stay flat in other regions.

As discussed in the Sect. 3.5, trends in recent years are more uncertain as they rely on sometimes preliminary activity data and emission factors extended outside inventory scaling years. Some of

the notable trends in CEDS emissions estimates in recent years are also from particularly uncertain sources. OC and BC emission estimates have some of the highest degrees of uncertainty in global inventories, and waste sectors in particular are highly uncertain. Additionally, a lot of global growth can be attributed to sectors that, in the CEDS system follow population trends over the most recent few years (e.g. waste, agriculture, and residential

biomass); are from inherently uncertain sectors (e.g. waste); or in China where emissions remain uncertain because the accounting of emissions factors, fuel properties, and energy use data have been subject to corrections and subsequent debate (Hong et al., 2017; Korsbakken et al., 2016; Liu et al., 2015b; Olivier et al., 2015).

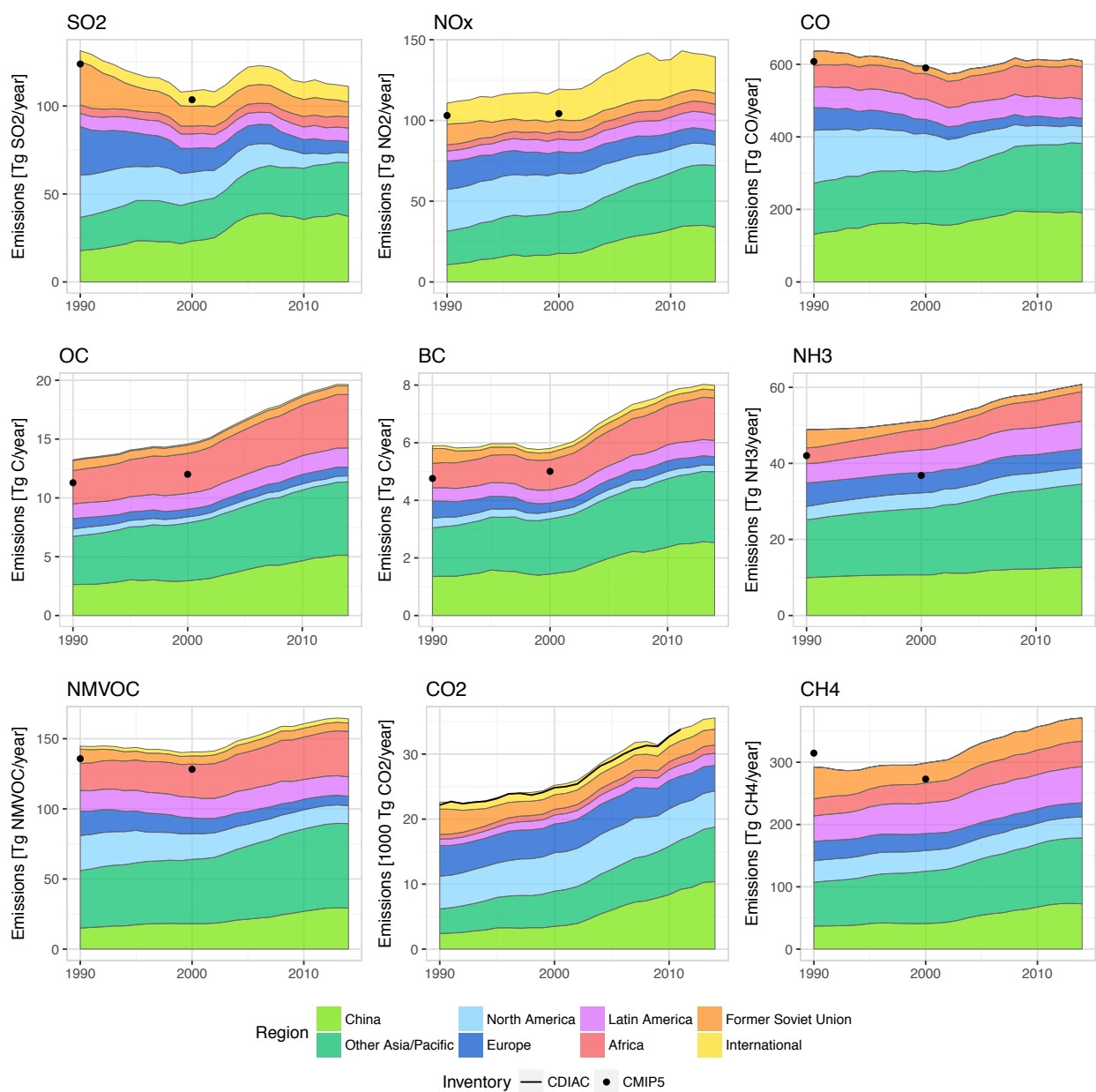

**Figure 5: Recent emissions estimates (1990 - 2014) by region compared to Lamarque et al. (2010) (dots) and CDIAC (line) for CO₂. Shows same data as Figure 3 over a shorter time scale. For like with like comparison, these figures do not include aviation or agricultural waste burning on fields. The "International" region shows international shipping emissions.**

## 3.3 Gridded Emissions

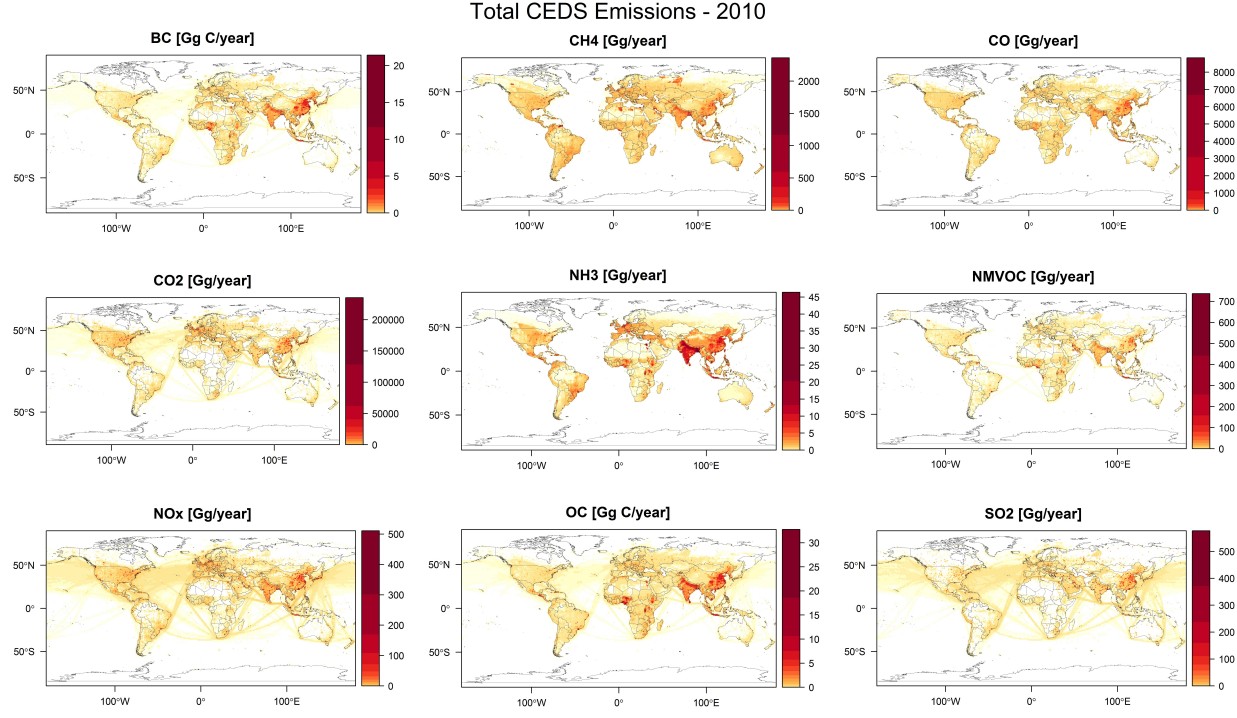

**Figure 6: Total gridded CEDS emissions by emission species for 2010**

Figure 6 shows gridded CEDS estimates of total emissions in 2010 for all emission species. CEDS maps are similar to existing maps such as EDGAR (EC-JRC/PBL, 2012) and CMIP5 (Lamarque et al., 2010) as these data sets are used in the gridding process. Emissions for most species are concentrated in high population areas such as parts of China, India, and the eastern US. BC and OC, whose emissions are dominated by heating and cooking fueled by biomass are also more concentrated in Africa. Shipping emissions are concentrated over along ocean shipping lanes for $NO_X$, $SO_2$, and $CO_2$. Discussion of how gridded data varies from CMIP5 (Lamarque et al., 2010) gridded data is included in Sect 3.4.1.

## 3.4    Comparison with other inventories

Differences between CEDS emissions and other inventory estimates are described below. The reasons depend on emissions species, but are largely due to updated emissions factors, increased detail in fuel and sector data, and a new estimate of waste emissions (however, see Sect. 3.5).

### 3.4.1   CMIP5 (Lamarque et al., 2010)

The emissions data used for CMIP5 (Lamarque et al., 2010) also used a "mosaic" methodology, combining emission estimates from different sources. The CEDS methodology provides a more consistent estimate over time since driver data are used to produce consistent trends. Emissions in earlier years, particularly before 1900, also differ because CEDS differentiates between biomass and coal combustion, which has a large impact on CO and $NO_X$ emissions. The (Lamarque et al., 2010) estimates for early years were drawn from the EDGAR-HYDE estimates (van Aardenne et al., 2001), which did not distinguish between these fuels. Figures showing comparisons between CMIP5 and CEDS globally by sector and for the top 5 emitting CMIP5 regions are shown in Sect. H of the Figures and Tables Supplement.

CEDS global $SO_2$ estimates are similar to CMIP5 estimates, although slightly lower (~10%) in the mid-20[th] century and slightly higher (~5%) near the end of the 20[th] century. Similar methods and data were used to develop both estimates (Smith et al., 2011). FSU $SO_2$ emissions are larger in CEDS (see Smith et al. 2011) from 1970-2000, but smaller in Europe from 1930 - 1980. Shipping $SO_2$ emissions are lower in the early 20[th] century due to updated methodologies (Smith et al. 2011), and slightly lower in recent years due to updated parameter estimates (see SI and Figure S43.

CEDS $NO_X$ emissions are smaller than the CMIP5 estimates until the mid-20[th] century. This is largely because of explicit representation of the lower $NO_X$ emissions from biomass fuels in early periods, which combusts at lower temperatures as compared to coal. In 1970 CEDS $NO_X$ emissions begin to diverge from CMIP5 estimates, generally larger due to waste, transportation, and energy sectors. CEDS emissions remain about 10% larger than CMIP5 in 1980 and 1990. Both global estimates increase and start to flatten around 1990. However, CEDS values flatten until 2000 and then increase again, while CMIP5 values decrease from 1990 to 2000.

CEDS CO estimates before 1960 are increasingly larger than CMIP5 estimates going back in time, reaching a factor of two by 1850 due to the explicit representation of biomass. In 1900, CEDS estimates are 70% larger than CMIP5, 98% of which is due to the RCO sector. CEDS estimates are slightly larger than CMIP5 post 1960 (8% in 1960 and 1970 and less than 5% from 1980 – 2000).

CEDS OC estimates are within 10% but smaller than CMIP5 estimates through 1970, when CEDS estimates quickly increase and become larger (at most 25% larger) than CMIP5 estimates. BC emissions are similar, although CEDS estimates are smaller (sometimes by 25%) than CMIP5 until 1960 when CEDS estimates increase quickly, up to 25% larger than CMIP5 estimates, in part due to larger waste sector emissions (see § 3.5). Differences in BC in the early 20th century are mostly from residential fuel use in the US. In 1910, 98% of the difference between the two inventories is from residential energy use, with 77% of that difference in the USA. US residential biomass consumption in 1949 is estimated using EIA data and propagated back in time to merge with Fernandes et al. (2007) used by SPEW in 1920. This US biomass estimate may be lower than that used in CMIP5.

$NH_3$ and NMVOC emissions are similar to CMIP5 estimates until 1950 when CEDS emissions begin to grow at a faster rate than CMIP5 emissions through 1990 when they are about 20-30% larger. Between 1990 and 2000 CMIP5 estimates show a decrease in emissions while CEDS estimates shows flattening emissions then a steep increase. Differences in NH3 emissions are largely due to steadily increasing agricultural emissions and a larger estimate from wastewater/human waste, which makes up 14% of CEDS $NH_3$ estimates in recent decades but was largely missing in the RCP estimates. CEDS NMVOC emissions are much larger for global waste, while much smaller for global transportation.

Global CEDS $CH_4$ emissions range from 93% of CMIP5 values in 1970 to 109% of CMIP5 values in 2000. CEDS estimates change more smoothly over time, without a dip in 2000. CEDS energy estimates are consistently larger than CMIP5 emissions, 22 – 58%, while CEDS agriculture emissions are consistently 10-15% smaller than CMIP5 estimates, except in 2000 (6% smaller) when CMIP5 estimates dip and CEDS emissions flatten, due to our inclusion of FAO agriculture data.

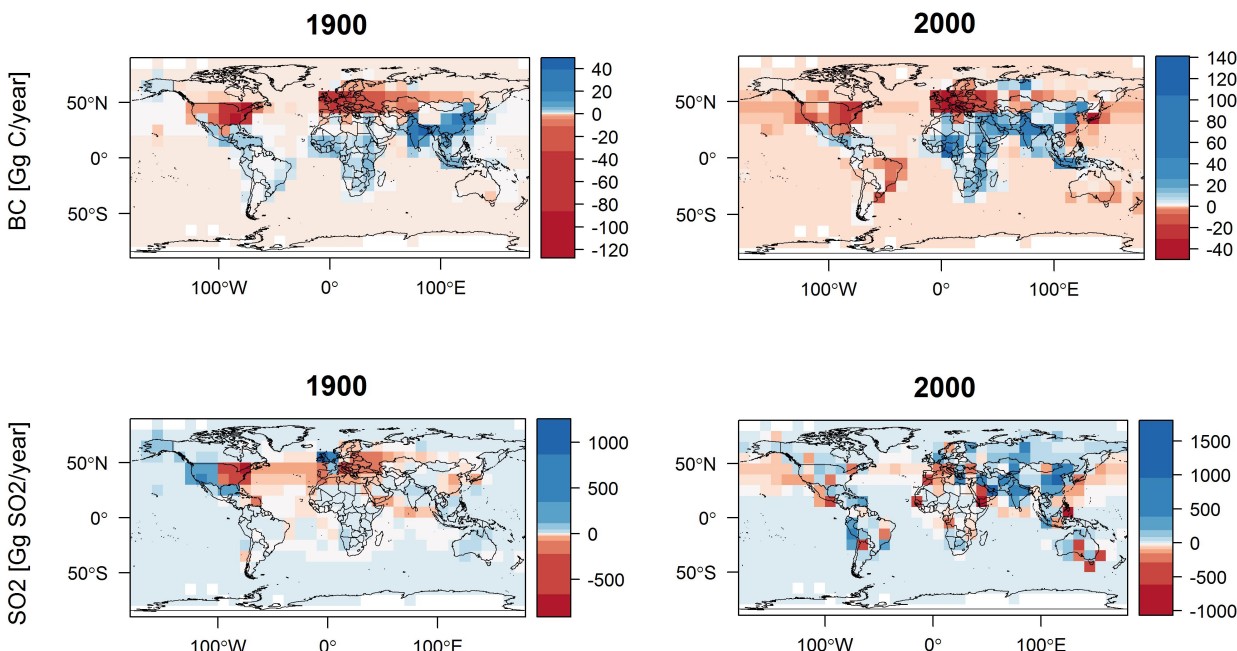

**Figure 7: Difference between CEDS and CMIP5 total gridded emissions for BC (top) and SO₂ (bottom) in 1900 (left) and 2000 (right) at 10 degree grid cells. Values shown are CEDS – CMIP5 estimates. For like with like comparison, these figures do not include aviation or agricultural waste burning on fields.**

Figure 7 shows differences between total gridded emissions for CEDS and CMIP5 for BC and SO₂ in 1900 and 2000. In 1900, CEDS BC emissions are lower over the US, Europe (especially cities in the UK) and larger over parts of India and China. Larger differences are concentrated in high population areas. In 2000, emissions follow a similar pattern. CEDS BC emissions are smaller over Europe and the eastern US, but larger over populated areas of India, China, and western Africa (particularly Nigeria), reflecting, in part higher country totals (e.g. Figure S41).

Additional text and similar difference maps for NO$_X$, CO, OC, NH₃, and NMVOC as well as high resolution figures for SO₂, are included in Supplemental Figures and Tables, Sect. K. The magnitude of most differences in 1850 are small, as total global emissions are small, and tend to be more concentrated in populated areas, with larger differences by 1900. Differences in 2000 are a bit larger, and tend to be consistent across countries. For example, total CEDS CO emission in India in 2000 are smaller than CMIP5 values, so most grid cells in India have smaller values.

However, differences in gridded SO₂ emissions in 2000 are not as consistent across countries or regions and tend to be highly concentrated into small groups of grid cells. Globally CEDS SO₂ emissions are very similar to CMIP5 emissions, and emissions are dominated by large point sources, so these differences are likely due to updated proxy data for power plants and metal smelters. The distribution of SO₂ emissions over the US also differ from CMIP5 grids, shown in Figure 7 and Sect. K of the Supplemental Figures and Tables, and detailed in Section A2.1.

BC, OC and NH₃ CEDS emissions in 2000 are larger over India, China, and parts of Africa than CMIP5 estimates, similar to BC emission in Fig. 7. CEDS NO$_X$ emissions in 2000 are also larger over China and India, while they are smaller over the Middle East and Eastern Europe. NMVOC estimates are smaller over China and the Middle East.

As discussed further in the SI (section K), these differences are due to a combination of differences in aggregate country-level emission estimates, spatial proxy data, and methodologies for mapping aggregate emissions to spatial grids. We note that the spatial proxy that is most important will also depend on emission species: for $SO_2$ power plants will generally be a key sector while for $NO_2$ mobile sources are an important sector over recent decades.

### 3.4.2  GAINS and EDGAR v4.3

CEDS estimates are compared to GAINS and EDGAR v4.3 emissions estimates in Fig. S40, shown in Supplemental Figures and Tables (SI-Figures).

Comparing GAINS with CEDS, BC, OC, $NO_X$, and $SO_2$ CEDS estimates are within +/- 20% of global GAINS values in 2000, 2005 and 2010. OC and $SO_2$ CEDS emissions are smaller than GAINS values in 2000, but become larger than GAINS global values by 2010. CEDS $NO_X$, $CO_2$, and BC emissions are consistently smaller than GAINS estimates and CEDS CO estimates are consistently larger than GAINS but within 6%, while CEDS NMVOC are 26 – 43% larger than GAINS estimates from 2000 - 2010.

BC emissions increase by about 10% from 2000 to 2010 in GAINS while the increase is 33% in CEDS. Two particularly large differences are due to coke production in China, which are particularly uncertain, and residential emissions from biofuel use (see Figure 4), both of which increase significantly over this period in CEDS.

Between 2000 and 2010 global CEDS emissions for all species (except $CO_2$) increase more than the GAINS estimates, with CEDS estimates are higher than GAINS by 2010 a number of species (Figure S40). GAINS emissions exhibit slower growth than CEDS emissions in recent years, indicating that GAINS includes more emissions controls or other changes over this period than CEDS (and the inventories to which CEDS is calibrated). The divergence in recent years, is particularly present in $SO_2$ and $NO_X$ emissions for power generation in China and India and $SO_2$ globally from refineries. This divergence continues out to 2015 (IEA, 2016b, based on an updated version of GAINS), in which global $SO_2$ emissions decline by ~25,000 Gg from 2005 to 2015, while CEDS emissions decline by only ~10,000Gg over 2005 to 2014.

CEDS estimates are consistently larger than EDGAR 4.3 global estimates for most emissions species. CEDS emissions follow the similar trends as EDGAR from 1970 – 2000 or all species but OC. CEDS emissions for OC grow somewhat linearly over the period, while EDGAR estimates stay relatively flat. Sectors driving the differences between CEDS and EDGAR estimates vary by emission species. However, these differences are largely due to waste burning and aggregate sector 1A4, which is dominated by residential emissions, but also includes commercial/institutional, and agriculture/forestry/fishing. A key differences is associated with estimates for waste (trash) burning which are much higher in CEDS (based on Wiedinmyer et al. (2014)) and have a strong influence on totals, particularly OC, with smaller relative impacts on NMVOC and BC (see §3.5).

Global CEDS $CH_4$ emissions estimates are slightly smaller than, but similar to EDGAR 4.2 estimates, ranging from 94 – 98% the EDGAR estimates. The similarity is because much of our methane emissions are either from EDGAR, or FAO (which uses similar methodologies). The largest differences can be found in 1B2 (fugitive petroleum and gas emissions) in Central and South America Africa, and the Former Soviet Union, as these default emissions also incorporate data from ECLIPSE V5a (Stohl et al., 2015), and rice cultivation in China (FAO, 2016).

### 3.5 Uncertainty

Emission uncertainty estimates in inventories are a critical need, however this is difficult to quantify and most inventories do not include uncertainty estimates. All the components and assumptions used in this analysis are uncertain to varying degrees, which means that uncertainty will vary with time, space, and emission species making quantification of uncertainties challenging.

There are some consistent trends in uncertainty estimates by emission species. Uncertainty is generally lowest for $CO_2$ and $SO_2$ emissions, which depend primarily on quality of fossil fuel statistical data and fuel properties, e.g. carbon and sulfur content, with straightforward stoichiometric relationships. Global $CO_2$ and $SO_2$ uncertainty has been estimated to be in order 8% for $CO_2$ (Andres et al., 2012) and 8-14% for $SO_2$ (Smith et al., 2011), for a roughly 5-95% confidence interval. Global uncertainties for these species tend to be relatively low also because fuel properties are not thought to be highly correlated between major emitting regions.

Uncertainty in specific countries can be much higher, however. China is a major emitter of both $CO_2$ and $SO_2$, and uncertainties regarding the level of coal consumption (Guan et al., 2012; Liu et al., 2015b) will directly impact emission estimates as well as actual implementation and efficiency of control equipment (Xu et al., 2009, 2009; Zhang et al., 2012). Since China energy consumption uncertainties appear to be largest in sectors with limited emission controls they can have a large impact on $SO_2$ emissions in particular (Hong et al., 2017). There is also uncertainty regarding the appropriate $CO_2$ emission factor for coal in China (Liu et al., 2015b; Olivier et al., 2015) as discussed further in the SI-Text Sect. 5.4.

Emission factors for CO, $NO_X$, NMVOC, BC and OC, tend to be dependent on details of the emitting process, and, therefore, have higher uncertainties (Blanco et al., 2014). This is particularly true for carbonaceous aerosol emissions, where emission factors can range over several orders of magnitude depending on the conditions under which combustion occurs. Uncertainties in global BC emissions have been estimated to be a factor of two (Bond et al., 2004). Uncertainty in country-level BC emissions in China were estimated to be −43% to +93% by Lu et al. (2011), -50% to +164% by Qin and Xie (2012) ±176% by Kurokawa et al. (2013a), and -28 to +126% by Zhao et al. (2013b). Uncertainty in activity levels also contributes. Solid biomass consumption is difficult to track and both absolute values and trends are generally much more uncertain than fossil fuel consumption data, which will contribute to BC and OC emissions uncertainty.

Emissions uncertainties for CO, $NO_X$, NMVOC typically lie between those of carbonaceous aerosols and those of $CO_2$ and $SO_2$. In part this is because, particularly in industrialized economies, a number of sectors contribute to emissions, and sectoral uncertainties will largely be independent of each other. Substantial uncertainty can still be present for specific sectors, even in countries with well-developed emission inventory processes (Parrish, 2006). For example, studies combining observations and modeling suggest that recent US national emissions inventory overestimates on road vehicle $NO_X$ emissions by about a factor of two (Anderson et al., 2014; Hassler et al., 2016; Travis et al., 2016), while recent updates of Canadian NMVOC emissions (Environment and Climate Change Canada, 2016) are, for some sectors, a factor of two larger than previous estimates (Environment Canada, 2013).

There are specific sectors with particularly uncertain emissions. The level of fugitive emissions often depends on procedures and practices, leading to large uncertainty. Emissions that result

from biological processes, such as $NO_X$ from fertilized soils or $NH_3$ from wastewater and agriculture, also generally depend on environmental conditions and would, in principle, require detailed modeling to improve estimates. Our $NH_3$ emissions from human waste, for example, adapt the methodologies used in REAS (Kurokawa et al., 2013a) and uses a single global default emission factor (modified to account for wastewater treatment as described in the SI). Not only is

this emission factor uncertain, but there will certainly be regional variations due to differing environmental conditions that we were unable to take into account. For agricultural emissions, the actual practices of managing livestock manures will affect true emissions; such practices vary significantly across the world but are not always well understood or reflected in the emission factors used in global inventories (Paulot et al., 2014). We note that in the CEDS historical

extrapolation before either 1960 or 1970, depending on the sector, global trends were used for agricultural emissions, which means that country-specific trends were not taken into account, leading to additional uncertainties at the country level.

Residential waste burning emissions depend on the amount of waste combusted, composition of the waste, and combustion conditions. This sector globally contributes a substantial fraction of

OC emissions in particular, but substantial amounts of BC and other species. The CEDS estimate for this sector, except where scaled to country emission estimates (available only in a few OECD countries) is based on 2010 estimates from Wiedinmyer et al. (2014). Wiedinmyer et al. followed IPCC guidelines and assumed that 60% of all waste that is not reported as collected is burnt. This could be an overestimate in countries where there is informal waste collection and recycling.

Klimont et al. (2017a) recently estimated BC and OC emissions from this sector, estimating that from 115 to 160 Tg of waste was openly burned, while Wiedinmyer et al. (2014) derive a value of 970 Tg. It is possible that the CEDS values, therefore, are overestimates of emissions from this source. Note, however, that the Wiedinmyer et al. (2014) estimate only includes residential waste burning. In the USA, for example, a large portion of CO2 from waste burning is from

industrial waste, particularly from tires (US EPA, 2015), which implies there may also be additional air pollutant emissions from industrial waste combustion. Outside of the specific OECD countries where country-specific inventories include this sector, industrial waste estimates were not explicitly included in the CEDS estimates. Overall there is substantial uncertainty for emissions from this sector.

All other factors being equal, uncertainty will tend to increase backwards in time, as driver data becomes more uncertain and older technologies are used, for which emission factors are not well quantified. We generally expect that uncertainty in this data set will be smaller for those years and countries where robust inventory development mechanisms are in place. However, as noted above for $NO_X$ in the USA, this does not eliminate uncertainty. Official country inventories can

sometimes be developed with outdated methodologies or can be incomplete. Many countries have regular evaluation activities, which indicate deficiencies and potential areas for improvement. However, assessments of completeness and plausibly are always useful, and inventories developed for scientific use, including CEDS, can help contribute in this area.

Our data system also allows us to examine the emission factors implied by scaling to country

inventories. This can reveal potential inconsistencies or regional differences. One example is shown in Figure 8, which shows the implied emission factor for CO emissions from gasoline road vehicles. Even where there is a mix of fuels in the road sector, the much higher CO emission factor for gasoline tends to lead to gasoline dominating emissions, making this comparison a fairly unambiguous reflection of underlying inventory assumptions. There is over a

factor of two difference in implied emission factors before 1990, with some inventories indicating steadily increasing emission factors going back in time while others flatten out. It is unclear if these differences are due to local variations in vehicle types, operation, or environmental conditions, or if differences reflect inventory assumptions, which implies some inventories might be biased high or low.

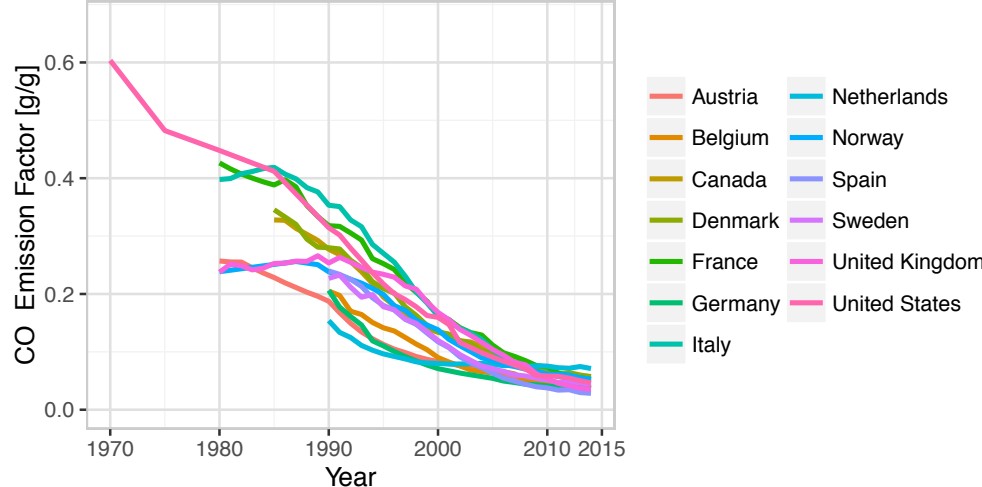


**Figure 8: Implied CO emissions factor for gasoline road vehicles obtained by the CEDS system after scaling to match country inventories. Data points only shown where an inventory value was available in units of g CO/g fuel.**

There are specific issues with uncertainty over the most recent few years in most emission data sets. We have, in this data set, provided emissions up to 2014. Emissions estimated for the most
recent several years are likely to have larger uncertainty due to the use of incomplete or preliminary data. Uncertainty in recent years comes from three main sources: activity data, emissions inventories that are used in our estimate, and the treatment of emission factors. Uncertainty from activity data comes from both uncertainty in country totals and their sector split. While activity data are often updated annually, recent estimates sometimes change for a
few years after their initial release. For example, the BP estimate of Russian coal use in 2012 may be different in the 2013, 2014, and 2015 data releases. The BP estimates we use to extrapolate fuel use for the most recent 2 years (Sect. 2.2.1) also lacks sectoral detail, which adds to uncertainty. Values in the inventory estimates we use in this data set for the most recent year are often preliminary and are later revised, which is an additional source of uncertainty.

Finally, we use emission factor trends from GAINS to project emission factors for combustion sectors for recent years beyond where inventory data are available. The last inventory year varies: 2010 for EDGAR, which is our default inventory for most species, 2008 for REAS, 2012 for China, 2013 for most of Europe, and 2014 for the USA. Using emissions factor trends that are not from detailed country-specific inventories is an additional source of uncertainty.

In future versions of CEDS, quantitative uncertainty analysis will be included for all time periods, as further discussed in Sect. 5.

## 4    Comparisons with observations

It is challenging to evaluate emissions against observations since, other than facility-specific emissions monitors, concentrations of emitted species are observed rather than emissions fluxes

930 into the atmosphere. Satellite data (Jacob et al., 2016; Streets et al., 2013), road-side measurements (Pant and Harrison, 2013), and inversion of surface observations (Bruhwiler et al., 2014; Houweling et al., 2017) can all be used to estimate emissions using observational data. These techniques can be used to gain insights into the accuracy of emission inventories, although each has associated uncertainties. Emission ratios are a particularly valuable technique, and we

935 compare in this section CEDS data with observations for two cases.

Hassler et al. (2016), compare observed ambient $NO_X$/CO enhancement ratios (measurements taken during morning rush hour) with $NO_X$/CO road emissions trends for London, Paris, and several US cities. Hassler et al. compare to the MACCity inventory (Granier et al., 2011), which is based on CMIP5 (Lamarque et al., 2010) inventory estimates and RCP projections. They find

940 that log linear trends in observed ratios in US cities, London, and Paris are steeper than MACCity ratios by a factor of 2.8 – 5.5. CEDS country-level $NO_X$/CO emissions ratios match observed trends much closer than MACCity, where observed trends are only 2-18% steeper than CEDS trends, shown in Table 7. Further, CEDS gridded road emissions, match even better with the observed trends for London and Paris.

945 **Table 7  Trends in Observed and Inventory $NO_X$/CO emission ratios**

| City/Country | Years | Observed* | MAACity* | CEDS (gridded - road) | CEDS aggregate (road) | CEDS (total) |
|---|---|---|---|---|---|---|
| USA (various cities) | 1989-2013 | 4.1 | 1.45 | | 3.86 | 2.37 |
| UK (London) | 1989-2015 | 7.2 | 1.88 | 6.92 | 6.90 | 5.90 |
| France (Paris) | 1995-2014 | 8.8 | 1.59 | 8.09 | 7.47 | 3.39 |
| Values shown in log linear trends in units of %yr$^{-1}$ *(Hassler et al., 2016) | | | | | | |

Kanaya et al. (2016) present observations of BC/CO ratios over six years (2009 – 2015) at Fukue Island, Japan, which, depending on wind conditions, gives region specific emission ratios under dry conditions for Japan, Korea, and four regions in China, shown in Table 8 compared to CEDS

950 and REAS BC/CO emissions ratios, both of which do not include open biomass burning. Both CEDS and REAS emissions ratios are similar to observed ratios for Japan, 1.64 and 1.1 times larger than observed ratios respectively, but near the observational uncertainty. The 2008 – 2015 average CEDS emission ratio is 2.1 – 2.7 times larger than observed ratios over China regions.

CEDS emissions ratios are substantially larger than both observed and REAS ratios for Korea.

955 Kanaya et al. attribute the difference between REAS and observations in Korea to the overestimation of industry and transportation BC/CO ratios in inventories. CEDS Korea, sector-specific BC/CO emissions ratios are high compared to observations: 370 and 41 ngm$^{-3}$ppb$^{-1}$ for industry and transportation sectors respectively compared to 42 and 27 ngm$^{-3}$ppb$^{-1}$ in REAS. CEDS CO estimates, which are scaled to Korean national inventory from 1999 – 2012, are 5 –

960 47% lower than REAS2.1 estimates over 2000 – 2008. CEDS CO emission estimates are dominated by energy transformation (20%) and transportation 68%. CEDS BC estimates use SPEW assumptions. CEDS BC emissions estimates for Korea are 5-8 times larger than REAS estimates. While CEDS estimates are larger over all sectors, the other transformation (e.g. coal coke production) and road sectors are the primarily sources. Emissions from the CEDS other

965 transformation sector, which are zero in REAS estimates, makes up 35% of CEDS Korea

estimates. CEDS Road BC emissions over 2000 - 2008 are 2-3 times larger than REAS estimates and 34% of the CEDS total.

These comparisons are approximate, given that the CEDS data represents entire countries and the air trajectories sampled at Fukue Island will preferentially sample only portions of each country. In future versions of CEDS we plan to produce emissions for large countries such as China at the province level which will aid in such comparisons. In general, differences in these ratios could be attributed to the overestimation of BC, underestimation of CO emissions, or both. Overall, CEDS emissions appear consistent for Japan, but perhaps slightly too high for China. CEDS BC estimates for Korea are quite high compared to other inventories and the observations, and suggest that the SPEW emission factors for Korea may not have incorporated the impact of transportation emission controls and new technologies for coal coke production.

**Table 8 Observed and Inventory BC/CO emission ratios**

| Country | Observed* 2009 - 2015 [ΔBC/ΔCO] | CEDS 2009 - 2014 [BC/CO] | CEDS 2008 [BC/CO] | REAS2.1 2008 [BC/CO] |
|---|---|---|---|---|
| Japan | 5.9 ± 3.4 | 9.7 | 9.5 | 6.5 |
| Korea | 6.7 ± 3.7 | 89.8 | 82.3 | 23 |
| China (North East) | 6.0 ± 2.8 | 14.3 | 12.8 | |
| China (North Central East) | 5.3 ± 2.1 | | | 8.3 |
| China (South Central East) | 6.4 ± 2.2 | | | 9.9 |
| China (South) | 6.9 ± 1.2 | | | |
| Values shown in ngm$^{-3}$ppb$^{-1}$ * (Kanaya et al., 2016) | | | | |

These examples illustrate that further comparisons would be of substantial value in better resolving emissions. The use of multiple observations and methodologies would add confidence to conclusions regarding the accuracy of emission inventory data.

## 5   Limitations and future work

While this data set includes many improvements upon existing comprehensive, long-term inventories, there are some specific limitations of the current methodology, and plans for improvement, that we discuss here.

Disaggregation of key non-combustion sectors, particularly 1A1bc_Other-transformation and 2C_Metal-production, should allow a more accurate estimation of emissions trends. This will require collection of additional activity data and default emission factors. At the current level of aggregation, emission trends for these sectors will be less accurate, particularly for years where country-level emission data sets are not available.

Emissions trends could be further improved for the mid-20th century. Emission factors here are often the result of scaling at later inventory years (e.g. Fig. 4), and further work to better constrain emission factors over this period is needed. The sectoral spilt for fuel use is also approximated over this period; incorporation of regional activity data would improve this as well. Non-combustion emissions are particularly uncertain in the era before modern inventory

data sets, which is generally before 1970/1980, since these emissions can depend on process details.

We plan to incorporate more detailed data from the US National Emissions Inventory, although as with the current estimate, discontinuities due to methodological changes will need to be addressed. Use of this data to estimate emissions at the US state level is underway, which will also be used to improve the spatial gridding of emissions over time.

Currently, a number of gridding proxies are static over time. Residential (and related) emissions are distributed using population distribution, which does change over time. Because residential emissions are dominant in earlier years, much of the major shifts in spatial distribution within countries are being captured. Other sectors have mix of spatial proxies, few of which are newer than 2010, and many were kept static over time. Shipping emissions patterns have changed over time, however we lack consistently constructed spatial proxies over time. The shipping spatial data used here (from ECLIPSE) has a higher fraction of emissions in the North Atlantic than the spatial distribution used in CMIP5. It is not clear if this difference is due to different methodologies or an actual change in spatial distribution over the last decade. Consistent data sets over time for spatial proxy information would be a useful addition.

A major next step in this project will be estimation of uncertainty. Our first step will be quantification of the additional uncertainty that stems from producing estimates out to the most recent full year, followed by comprehensive uncertainty estimates that will be used to produce ensembles of emissions to more fully reflect the uncertainty in these data.

In addition to updates, refinements, and uncertainty analysis, the CEDS system will be released as open-source software, along with associated input data. Where previous work has only released final emissions estimates, this entire data system will be released to facilitate evaluation of trends in and the relationships between emissions, emission factors, and their drivers across time, countries, sectors, and fuels; to foster transparency in assumption and methods; and allow community input and participation. While the current data system requires purchase of the IEA energy statistics, we will explore options to facilitate use with publically available data as well.

## 6    Summary

This paper described the methodology and results for a new annual data set of historical anthropogenic GHGs, reactive gases, aerosols, and aerosol and ozone precursor compounds from 1750 to 2014 for use in CMIP6. This data set relies heavily on IEA energy statistics, EDGAR, and other inventory data sets to produce consistent trends over time. Key steps in estimating emission include collecting existing activity, emissions factors, and emissions data; developing default emissions estimates; calibrating default estimates to existing inventories; extending present day emission to historical time periods; and gridding emissions.

Emissions before 1850 are dominated by residential biomass burning and agricultural emissions. As the industrial revolution expanded, energy, industry, and transportation related emissions then begin to grow and then quickly increase in the mid 20[th] century. Emissions of some species begin to slow or see global reductions in the late 20[th] century with the introduction of emission control policies, but emissions of many of those species increase again in recent years due to increased economic activity in rapidly industrializing regions. While comparable to existing data sets such as CMIP5 (Lamarque et al., 2010), EDGAR (EC-JRC/PBL, 2016), and GAINS (Amann et al.,

2011; Klimont et al., 2017a), CEDS estimates are generally slightly higher than those inventories in recent years.

Future work on this data system will involve refining and updating these emissions estimates, adding detail, and the release of the CEDS as an open source data system. In order to be able to release the current data set in time for use in CMIP6, the focus was on the development and use of a consistent methodology, relying in large part on IEA energy statistics and existing inventory data over recent years. As described above and in the SI, a number of additions were made

where inconsistencies or incompleteness in these core data sets were known and improved data were readily available. There are many further corrections that would likely be useful to implement. For example, the inventories used here for calibration may already be known to contain deficiencies, for example through regular validation activities. There are likely also country level energy and other driver data that can be used to improve the data used here.

Finally, further detailed comparisons with observations may help to indicate additional areas where changes to emission factor or other assumptions are warranted.

With the release of this data set, and soon the entire data system, it is our intention that further improvements will be made through feedback from the global emissions inventory community. The CEDS data system, including R code and all input data other than the IEA energy statistics,

is being prepared for public release in fall 2017 through the gitHub collaboration website. This will facilitate community comment, and direct contributions to improving these emissions data. The next data release is planned for Fall/Winter 2017, which will extend the time series to 2016 and correct, to the extent possible, any known issues with the dataset. We aim to continue annual updates in subsequent years. We welcome comments, including notes on any

potential inconsistencies or relevant new data sources, so that these data can be improved in future releases.

**Data Availability**

Gridded versions of this data are available through the Earth System Grid Federation (ESGF) [https://esgf-node.llnl.gov/search/input4mips/] under the activity_id = "input4MIPs" and

institution = "PNNL-JGCRI". More information on the CEDS project, system release, and updates, can be found at http://www.globalchange.umd.edu/ceds/. Note that known issues with the data are listed at https://github.com/JGCRI/CEDS and users can also submit issues via the GitHub site.

**Author Contributions:**

R.M. Hoesly and S.J. Smith prepared the manuscript with contributions from L. Feng, Z. Klimont, G. Janssens-Maenhout, L. Vu, R. Andres, M.C.P. Moura, L. Liu, Z. Lu, and Q. Zhang. The CEDS system was developed by R.M. Hoesly, S.J. Smith, L. Feng, T. Pitkanen, J.J. Seibert, L. Vu, and R. Bolt. Analysis was performed by R.M. Hoesly, S.J. Smith, L. Feng, L. Vu, M.C.P.

Moura, N. Kholod, and P. O'Rourke. Data were contributed by Z. Klimont, G. Janssens-Maenhout, R.J. Andres, T.C. Bond, L. Dawidowski, J. Kurokawa, M. Li, L. Liu, Z. Lu, and Q. Zhang.

**Competing Interests**

The authors declare that they have no conflict of interest.

**Acknowledgements**

This research was based on work supported by the U.S. Department of Energy (DOE), Office of Science, Biological and Environmental Research as part of the Earth System Modeling program. Additional support for the development of the gridded data algorithm was from the National Atmospheric and Space Administration's Atmospheric Composition: Modeling and Analysis
Program (ACMAP), award NNH15AZ64I. The Pacific Northwest National Laboratory is operated for DOE by Battelle Memorial Institute under contract DE-AC05-76RLO1830. RJA was sponsored by U.S. Department of Energy, Office of Science, Biological and Environmental Research (BER) programs and performed at Oak Ridge National Laboratory (ORNL) under U.S. Department of Energy contract DE-AC05-00OR22725.

The authors would like to acknowledge Grace Duke and Han Chen for data collection and processing; Alison Delgado, Minji Joeng, and Bo Liu for data collection and translation; and Benjamin Bond-Lamberty and Robert Link for code review. We thank Kostas Tsigaridis for pointing out a discontinuity due to a data anomaly (spurious brown coal consumption over 1971-1984 in the IEA Other Asia region) in a review version of the inventory data.

**A1 Supplementary information files**

Supplementary files related to this article include:

Supplemental Data and Assumptions Text (pdf)

Supplemental Figures and Tables (pdf)

Data Files (zipped set of csv files)

- Emissions by country and sector (all species)
- Global emissions by sector (all species)
- Total emissions by country (all species)
- Country mapping and ISO codes

The supplementary information for this article describes a number of additional data sources
used in this work, including the following:

(Bartoňová, 2015; Blumberg et al., 2003; Denier van der Gon et al., 2015; EIA, 2013; Endresen et al., 2007; Environment Canada, 2016; Eyring et al., 2005; Fletcher, 1997; Foell et al., 1995; Fouquet and Pearson, 1998; Gschwandtner et al., 1986; Huo et al., 2012; IEA, 2016a; IMO, 2014; Kaur et al., 2012; Kholod and Evans, 2015; Liu et al., 2015a; Ludek and Holub, 2009;
McLinden et al., 2016; Mester, 2000; Mitchell, 2003, 2007, 1983; Mylona, 1996; OECD, 2016; Pretorius et al., 2015; Rowe and Morrison, 1999; Ryaboshapko et al., 1996; Sanger, 1997; Simachaya, 2015; Smith et al., 2014; Tushingham, 1996; UK DEFRA, 2015; US EPA, 2012a; Wu et al., 2012; Zhou et al., 2011)

**A2  Data Release and Known Issues**
**A2.1   Known Issues**

This section lists known issues with the data released as of this writing (August 2017). Readers should refer to the project web site for general updates (globalchange.umd.edu/CEDS) and the project's gitHub site for an updated list of issues (https://github.com/JGCRI/CEDS)..

- Combustion emissions become zero in earlier years for several countries that have
inconsistent temporal coverage in the IEA energy data. These include: Sint Maarten,

Suriname, Cambodia, Mongolia, Palau, Botswana, Namibia, and Niger. Some of these instances, where alternative data sources are available, will be corrected in the next release.

- Some of the countries in the IEA "other" aggregations (e.g. "Other Asia", "Other Africa", and "Other Non-OECD Americas") have spurious sector splits due to the simple methods used to assign fuel use to these countries (e.g. there is fuel use in the Afghanistan international shipping sector).

- There are a few spurious small-magnitude process emissions (particularly in 2C_Metal-production) for smaller countries before 1900 that are artifacts of the extension process. These have negligible impacts on emission totals.

- There are some spurious emission results for early years at the sectoral level in the current database due to the sectoral resolution of the data used to extend emissions back in time. For example, aircraft emissions are present back to 1851, even though actual aircraft emissions did not begin until the early 20th century. The magnitude of these emissions are small and, while these emissions should be zero in early years, these small magnitudes will not materially impact climate model results. ($NO_X$ in the CEDS aviation sector in 1920 is 0.2% of estimated $NO_X$ from lighting (Schumann and Huntrieser 2007), for example, and very much smaller in earlier years.) The historical energy code is being revised to be more flexible to improve our ability to incorporate additional energy data sets including, for example, historical estimates of aircraft fuel consumption.

- Due to an error, $SO_2$ emissions in the US are overestimated from about 1961 to 1969. The overestimate averages 22% over this period. This has been corrected for inclusion in the next data release. The previous and corrected time series is shown in the supplement.

- $SO_2$ emissions in the gridded data are overestimated in the western United States relative to the eastern United States. This spatial allocation is present in the EDGAR emission grids used for spatial mapping within each country.

### A2.2 Gridded Data Release History

There have been several releases of the CEDS gridded data. The underlying emissions by country, sector and fuel have been identical in all of these releases, as are total emissions by country and gridding sector (with the exception of small changes in 1850 emissions noted below).

**v2016-05-20**: Pre-industrial 1750-1850 data release

**v2016-06-18**: 1851 – 2014 data

**v2016-06-18-sectorDim**: Re-release of both preindustrial and 1851 – 2014 in a new netCDF format with sectors as an additional dimension in the data variable. This reformatting was necessary due to a limitation that was discovered within the ESGF system summer 2016. The reformatted data were released early Fall 2016

**v2017-05-18**: Re-release of entire dataset in order to correct two gridding errors discovered by users. 1) Inconsistent emission allocation to spatial grids within countries that resulted in incorrect spatial allocations and some large discontinuities in the gridded data. These issues were particularly apparent in spatially large countries such as the USA and China. 2) Minor inconsistencies in seasonal allocation, resulting largely in emissions that were too high in February. Total annual emissions within each country were not impacted by either of these issues.

Emissions are also fully consistent across 1850 in this release. There were small discontinuities in 1850 between the CEDS CMIP6 preindustrial release (v2016-06-18) and the later full CEDS release (v2016-07-26) due to updates in the data system. These differences are 0.5% for all species (except NMVOC which reaches 1.5%). In absolute terms these differences are very small (relative to, for example, open biomass burning emissions) and will not have a significant impact on simulation results.

A link to further examination of these issues, including comparison maps and time series comparisons, can be found at the project web site (globalchange.umd.edu/CEDS).

### A2.3 Methane Historical Extension

As several modeling groups participating in CMIP6 requested $CH_4$ emissions from 1850. We were not able to extend the consistent CH4 time series before 1970 due to the additional data that would need to be collected and processed. We have, however, produced a "rough cut" supplementary extension of $CH_4$ emissions from 1850 – 1970 by scaling with CMIP5 historical $CH_4$ estimates (Lamarque et al. 2010). These estimates were generated by scaling the CEDS 1970 estimates with the CMIP5 trends (ie: shifting CMIP5 trends to match CEDS values in 1970) by aggregate sector and the 26 sub-region level of the CMIP5 data. While these emission estimates are not fully consistent with the other CEDS emissions, they provide a longer time series, albeit with some additional uncertainty, for groups that would like to have these trends. These data are available as supplementary gridded information for CEDS version 2017-05-18 data through ESGF (see Data Availability section).

Biases in this extended dataset have already been identified. The waste sector is 30% of total anthropogenic $CH_4$ emissions by 1850. This is likely because earlier CMIP5 data are scaled back in time with population data. This is an overestimate of anthropogenic $CH_4$ emissions from this source at that time since landfills and wastewater treatment plants, which create the anaerobic conditions conducive to $CH_4$ emissions, did not start to come into widespread use until around 1930. However, as noted in the main paper, earlier CMIP5 emission estimates did not distinguish between biomass and coal combustion. $CH_4$ emissions from biomass combustion are much larger than those from coal combustion, which means methane emissions from the residential sector are underestimated in this extrapolation. A rough estimate indicates that these two effects are of similar (and offsetting) magnitude. Further work is necessary to better refine historical $CH_4$ emissions.

### A3 Sector definitions
### A3.1 Combustion emissions

Fuel combustion emission sectors in CEDS are defined in reference to corresponding IEA energy statistics energy flows as given in this table. One exception is evaporative emissions from road

transport, which are mapped to the 1A3b road transport sector, following general air pollutant inventory practice, even though this is a non-combustion emissions source. Also NMVOC evaporative emissions from oil tanker loading are not combustion emissions, but are categorized together with international shipping emissions.

Note that the current calibration (e.g., scaling) to country emission inventories is generally not performed at this level of detail, which means that sectoral emission values are more reliable at the aggregate sector level.

**Table A1 Sector Definitions of Combustion Emissions (IEA and NFR14 Codes)**

| IEA Energy Statistics | IEA Name | NFR14 Code | CEDS Working Sector Name | Aggregate Sector (Gridding) |
|---|---|---|---|---|
| MAINELEC | Main-Activity-Producer-Electricity-Plants | 1A1a | 1A1a_Electricity-public | Power_and_Heat |
| AUTOELEC | Autoproducer-Electricity-Plants | 1A1a | 1A1a_Electricity-autoproducer | Industrial_Combustion |
| MAINCHP | Main-Activity-Producer-CHP-Plants | 1A1a | 1A1a_Electricity-public | Power_and_Heat |
| AUTOCHP | Autoproducer-CHP-Plants | 1A1a | 1A1a_Electricity-autoproducer | Industrial_Combustion |
| MAINHEAT | Main-Activity-Producer-Heat-Plants | 1A1a | 1A1a_Heat-production | Power_and_Heat |
| AUTOHEAT | Autoproducer-Heat-Plants | 1A1a | 1A1a_Heat-production | Power_and_Heat |
| IRONSTL | Iron-and-Steel | 1A2a | 1A2a_Ind-Comb-Iron-steel | Industrial_Combustion |
| NONFERR | Non-Ferrous-Metals | 1A2b | 1A2b_Ind-Comb-Non-ferrous-metals | Industrial_Combustion |
| CHEMICAL | Chemical-and-Petrochemical | 1A2c | 1A2c_Ind-Comb-Chemicals | Industrial_Combustion |
| PAPERPRO | Paper,-Pulp-and-Print | 1A2d | 1A2d_Ind-Comb-Pulp-paper | Industrial_Combustion |
| FOODPRO | Food-and-Tobacco | 1A2e | 1A2e_Ind-Comb-Food-tobacco | Industrial_Combustion |
| NONMET | Non-Metallic-Minerals | 1A2f | 1A2f_Ind-Comb-Non-metalic-minerals | Industrial_Combustion |
| CONSTRUC | Construction | 1A2g | 1A2g_Ind-Comb-Construction | Industrial_Combustion |
| TRANSEQ | Transport-Equipment | 1A2g | 1A2g_Ind-Comb-transpequip | Industrial_Combustion |
| MACHINE | Machinery | 1A2g | 1A2g_Ind-Comb-machinery | Industrial_Combustion |
| MINING | Mining-and-Quarrying | 1A2g | 1A2g_Ind-Comb-mining-quarrying | Industrial_Combustion |
| WOODPRO | Wood-and-Wood-Products | 1A2g | 1A2g_Ind-Comb-wood-products | Industrial_Combustion |
| TEXTILES | Textile-and-Leather | 1A2g | 1A2g_Ind-Comb-textile-leather | Industrial_Combustion |
| INONSPEC | Non-specified-(Industry) | 1A2g | 1A2g_Ind-Comb-other | Industrial_Combustion |
| WORLDAV | World-Aviation-Bunkers | 1A3ai | 1A3ai_International-aviation | Aviation |
| DOMESAIR | Domestic-Aviation | 1A3aii | 1A3aii_Domestic-aviation | Aviation |
| ROAD | Road | 1A3b | 1A3b_Road | Road |
| * NA | Evaporative emissions from road transport | 1A3b | 1A3b_Road | Road |
| RAIL | Rail | 1A3c | 1A3c_Rail | Other_Surface_Transport |
| WORLDMAR | World-Marine-Bunkers | 1A3di | 1A3di_International-shipping | International-Shipping |
| * NA | Evaporative emissions from tanker loading | 1A3di | 1A3di_Oil_tanker_loading | International-Shipping |
| DOMESNAV | Domestic-Navigation | 1A3dii | 1A3dii_Domestic-navigation | Other_Surface_Transport |

| | | | (shipping) | |
|---|---|---|---|---|
| PIPELINE | Pipeline-Transport | 1A3ei | 1A3eii_Other-transp | Other_Surface_Transport |
| TRNONSPE | Non-specified-(Transport) | 1A3eii | 1A3eii_Other-transp | Other_Surface_Transport |
| COMMPUB | Commercial-and-Public-Services | 1A4a | 1A4a_Commercial-institutional | Residential_Commercial_Other |
| RESIDENT | Residential | 1A4b | 1A4b_Residential | Residential_Commercial_Other |
| AGRICULT | Agriculture/Forestry | 1A4c | 1A4c_Agriculture-forestry-fishing | Residential_Commercial_Other |
| FISHING | Fishing | 1A4c | 1A4c_Agriculture-forestry-fishing | Residential_Commercial_Other |
| ONONSPEC | Non-specified-(Other) | 1A5 | 1A5_Other-unspecified | Residential_Commercial_Other |


## A3.2 Non-combustion emissions

Non-combustion emission sectors (also generally referred to as process emissions in CEDS documentation) are defined in reference to corresponding EDGAR categories as given in this table. Note that the 1A1bc sector is actually combustion-related emissions, however this sector is
processed the same as non-combustion emissions in CEDS (see Sec 2.3.2).

**Table A2 Sector Definitions of Non Combustion Emissions (drawn from EDGAR Processes)**

| EDGAR Process Description | CEDS-Working-Sector-Name | Aggregate Sector |
|---|---|---|
| Fuel combustion petroleum refineries | 1A1bc_Other-transformation | Industrial_Combustion |
| coal mines | 1A1bc_Other-transformation | Industrial_Combustion |
| Fuel combustion BKB plants | 1A1bc_Other-transformation | Industrial_Combustion |
| Fuel combustion blast furnaces | 1A1bc_Other-transformation | Industrial_Combustion |
| Fuel combustion charcoal production plants | 1A1bc_Other-transformation | Industrial_Combustion |
| Fuel combustion coal liquefaction plants | 1A1bc_Other-transformation | Industrial_Combustion |
| Fuel combustion coke ovens | 1A1bc_Other-transformation | Industrial_Combustion |
| Fuel combustion gasification plants for biogas | 1A1bc_Other-transformation | Industrial_Combustion |
| Fuel combustion Liquefaction/Regasification | 1A1bc_Other-transformation | Industrial_Combustion |
| Fuel combustion non-specified transformation | 1A1bc_Other-transformation | Industrial_Combustion |
| Fuel combustion oil and gas extraction | 1A1bc_Other-transformation | Industrial_Combustion |
| Fuel combustion patent fuel plants | 1A1bc_Other-transformation | Industrial_Combustion |
| Gas works | 1A1bc_Other-transformation | Industrial_Combustion |
| BKB plants | 1B1_Fugitive-solid-fuels | Fugitive_Energy_Emissions |
| Fuel transformation coal liquefaction plants | 1B1_Fugitive-solid-fuels | Fugitive_Energy_Emissions |
| Fuel transformation patent fuel plants | 1B1_Fugitive-solid-fuels | Fugitive_Energy_Emissions |
| Production of brown coal | 1B1_Fugitive-solid-fuels | Fugitive_Energy_Emissions |
| Production of hard coal | 1B1_Fugitive-solid-fuels | Fugitive_Energy_Emissions |
| Production of peat | 1B1_Fugitive-solid-fuels | Fugitive_Energy_Emissions |
| Fuel transformation charcoal production plants | 1B1_Fugitive-solid-fuels | Fugitive_Energy_Emissions |
| Fuel transformation coke ovens | 1B1_Fugitive-solid-fuels | Fugitive_Energy_Emissions |
| Fuel transformation in gas works | 1B1_Fugitive-solid-fuels | Fugitive_Energy_Emissions |
| Chemical heat for electricity production | 1B2_Fugitive-petr-and-gas | Fugitive_Energy_Emissions |
| For blended natural gas | 1B2_Fugitive-petr-and-gas | Fugitive_Energy_Emissions |
| Fuel transformation gasification plants for biogas | 1B2_Fugitive-petr-and-gas | Fugitive_Energy_Emissions |
| Fuel transformation Liquefaction/Regasification pl | 1B2_Fugitive-petr-and-gas | Fugitive_Energy_Emissions |
| Gas-to-liquids (GTL) plants | 1B2_Fugitive-petr-and-gas | Fugitive_Energy_Emissions |
| Non specified transformation activity | 1B2_Fugitive-petr-and-gas | Fugitive_Energy_Emissions |
| Petrochemical industry | 1B2_Fugitive-petr-and-gas | Fugitive_Energy_Emissions |
| Transformation in Gas to liquids plants | 1B2_Fugitive-petr-and-gas | Fugitive_Energy_Emissions |
| Fuel transformation petroleum refineries | 1B2_Fugitive-petr-and-gas | Fugitive_Energy_Emissions |
| Production of oil | 1B2_Fugitive-petr-and-gas | Fugitive_Energy_Emissions |
| Production of gas | 1B2_Fugitive-petr-and-gas | Fugitive_Energy_Emissions |
| Production of oil | 1B2_Fugitive-petr-and-gas | Fugitive_Energy_Emissions |

| (None) | 1B2d_Fugitive-other-energy | Fugitive_Energy_Emissions |
|---|---|---|
| Cement production | 2A1_Cement-production | Minerals |
| Lime production | 2A2_Lime-production | Minerals |
| Lime production | 2A2_Lime-production | Minerals |
| Soda ash production and use | 2Ax_Other-minerals | Minerals |
| Brick production | 2Ax_Other-minerals | Minerals |
| Glass bottles | 2Ax_Other-minerals | Minerals |
| Glass production | 2Ax_Other-minerals | Minerals |
| Other non-metallic minerals | 2Ax_Other-minerals | Minerals |
| Other uses of carbonate | 2Ax_Other-minerals | Minerals |
| Ammonia production | 2B_Chemical-industry | Chemical-industry |
| Bulk chemicals production | 2B_Chemical-industry | Chemical-industry |
| Nitric acid production | 2B_Chemical-industry | Chemical-industry |
| Adipic acid production | 2B_Chemical-industry | Chemical-industry |
| Silicon carbide production | 2B_Chemical-industry | Chemical-industry |
| Calcium carbide production | 2B_Chemical-industry | Chemical-industry |
| Bulk chemicals production | 2B_Chemical-industry | Chemical-industry |
| Caprolactam production | 2B_Chemical-industry | Chemical-industry |
| Bulk chemicals production | 2B_Chemical-industry | Chemical-industry |
| N-fertilizer production | 2B_Chemical-industry | Chemical-industry |
| Specialities production | 2B_Chemical-industry | Chemical-industry |
| Sulphuric acid production | 2B_Chemical-industry | Chemical-industry |
| Titanium oxide production | 2B_Chemical-industry | Chemical-industry |
| Bulk chemicals production | 2B_Chemical-industry | Chemical-industry |
| Glyoxal production | 2B_Chemical-industry | Chemical-industry |
| Glyoxylic acid production | 2B_Chemical-industry | Chemical-industry |
| Crude steel production | 2C_Metal-production | Metals-industry |
| Blast furnaces | 2C_Metal-production | Metals-industry |
| Pig iron production | 2C_Metal-production | Metals-industry |
| Sinter production | 2C_Metal-production | Metals-industry |
| Pellet production | 2C_Metal-production | Metals-industry |
| Steel casting | 2C_Metal-production | Metals-industry |
| Ferro Alloy production | 2C_Metal-production | Metals-industry |
| Aluminium production | 2C_Metal-production | Metals-industry |
| Magnesium production | 2C_Metal-production | Metals-industry |
| Aluminium production | 2C_Metal-production | Metals-industry |
| Other non-ferrous production | 2C_Metal-production | Metals-industry |
| Gold production | 2C_Metal-production | Metals-industry |
| Copper production | 2C_Metal-production | Metals-industry |
| Mercury production | 2C_Metal-production | Metals-industry |
| Other non-ferrous production | 2C_Metal-production | Metals-industry |
| Lead production | 2C_Metal-production | Metals-industry |
| Other non-ferrous production | 2C_Metal-production | Metals-industry |
| Magnesium production | 2C_Metal-production | Metals-industry |
| Zinc production | 2C_Metal-production | Metals-industry |
| Paper production | 2H_Pulp-and-paper-food-beverage- | Pulp-and-paper-food- |
| Wood pulp production | 2H_Pulp-and-paper-food-beverage- | Pulp-and-paper-food- |
| Beer production | 2H_Pulp-and-paper-food-beverage- | Pulp-and-paper-food- |
| Bread production | 2H_Pulp-and-paper-food-beverage- | Pulp-and-paper-food- |
| Other food production | 2H_Pulp-and-paper-food-beverage- | Pulp-and-paper-food- |
| Wine production | 2H_Pulp-and-paper-food-beverage- | Pulp-and-paper-food- |
| Non energy use in petrochemical industry | 2L_Other-process-emissions* | Other_Non-Combustion |
| Non energy use in industry, transformation industr | 2L_Other-process-emissions* | Other_Non-Combustion |
| Non energy use in transport sector | 2L_Other-process-emissions* | Other_Non-Combustion |
| Other non energy use | 2L_Other-process-emissions* | Other_Non-Combustion |
| Non energy use in petrochemical industry | 2L_Other-process-emissions* | Other_Non-Combustion |
| Non energy use in industry, transformation industr | 2L_Other-process-emissions* | Other_Non-Combustion |
| Non energy use in transport sector | 2L_Other-process-emissions* | Other_Non-Combustion |
| Other non energy use | 2L_Other-process-emissions* | Other_Non-Combustion |
| Other non-combustion not elsewhere (NOT | 2L_Other-process-emissions* | Other_Non-Combustion |
| Solvents in glues and adhesives | 2D_Paint-application | Solvents |
| Solvents in graphic arts | 2D_Paint-application | Solvents |
| Solvents in paint | 2D_Paint-application | Solvents |
| Solvents in dry cleaning | 2D_Degreasing-Cleaning | Solvents |

| | | |
|---|---|---|
| Solvents in households products | 2D_Degreasing-Cleaning | Solvents |
| Solvents in industrial degreasing | 2D_Degreasing-Cleaning | Solvents |
| Solvents in chemical industry | 2D_Chemical-products-manufacture-processing | Solvents |
| Other solvents use | 2D_Other-product-use | Solvents |
| Prodcution and use of other products | 2D_Other-product-use | Solvents |
| Use of N2O as anesthesia | 2D_Other-product-use | Solvents |
| Solvents in leather production | 2D_Other-product-use | Solvents |
| Solvents in pesticides | 2D_Other-product-use | Solvents |
| Solvents in rubber and plastics industry | 2D_Other-product-use | Solvents |
| Solvents in vegetative oil extraction | 2D_Other-product-use | Solvents |
| Enteric fermentation by cattle | 3E_Enteric-fermentation | Agriculture_non-combustion |
| Enteric fermentation by buffalo | 3E_Enteric-fermentation | Agriculture_non-combustion |
| Enteric fermentation by sheep | 3E_Enteric-fermentation | Agriculture_non-combustion |
| Enteric fermentation by goats | 3E_Enteric-fermentation | Agriculture_non-combustion |
| Enteric fermentation by camels | 3E_Enteric-fermentation | Agriculture_non-combustion |
| Enteric fermentation by horses | 3E_Enteric-fermentation | Agriculture_non-combustion |
| Enteric fermentation by asses | 3E_Enteric-fermentation | Agriculture_non-combustion |
| Enteric fermentation by swine | 3E_Enteric-fermentation | Agriculture_non-combustion |
| Manure management of cattle | 3B_Manure-management | Agriculture_non-combustion |
| Manure management of buffalo | 3B_Manure-management | Agriculture_non-combustion |
| Manure management of sheep | 3B_Manure-management | Agriculture_non-combustion |
| Manure management of geese | 3B_Manure-management | Agriculture_non-combustion |
| Manure management of goats | 3B_Manure-management | Agriculture_non-combustion |
| Manure management of camels | 3B_Manure-management | Agriculture_non-combustion |
| Manure management of horses | 3B_Manure-management | Agriculture_non-combustion |
| Manure management of assess | 3B_Manure-management | Agriculture_non-combustion |
| Manure management of swine | 3B_Manure-management | Agriculture_non-combustion |
| Manure management of chicken | 3B_Manure-management | Agriculture_non-combustion |
| Manure management of ducks | 3B_Manure-management | Agriculture_non-combustion |
| Manure management of turkey | 3B_Manure-management | Agriculture_non-combustion |
| Separate category for Rice CH4 emissions (not in | 3D_Rice-Cultivation | Agriculture_non-combustion |
| Agricultural soils, rice cultivation | 3D_Soil-emissions | Agriculture_non-combustion |
| Agricultural soils, nitrogen fertilizers | 3D_Soil-emissions | Agriculture_non-combustion |
| Agricultural soils, animal waste as fertiliser | 3D_Soil-emissions | Agriculture_non-combustion |
| Agricultural soils, N-fixing crops | 3D_Soil-emissions | Agriculture_non-combustion |
| Agricultural soils, crop residues | 3D_Soil-emissions | Agriculture_non-combustion |
| Agricultural soils, histosols | 3D_Soil-emissions | Agriculture_non-combustion |
| Agricultural soils, buffalos in pasture | 3D_Soil-emissions | Agriculture_non-combustion |
| Agricultural soils, camels in pasture | 3D_Soil-emissions | Agriculture_non-combustion |
| Agricultural soils, cattle in pasture | 3D_Soil-emissions | Agriculture_non-combustion |
| Agricultural soils, chicken in pasture | 3D_Soil-emissions | Agriculture_non-combustion |
| Agricultural soils, ducks in pasture | 3D_Soil-emissions | Agriculture_non-combustion |
| Agricultural soils, goats in pasture | 3D_Soil-emissions | Agriculture_non-combustion |
| Agricultural soils, horses in pasture | 3D_Soil-emissions | Agriculture_non-combustion |
| Agricultural soils, mules and asses in pasture | 3D_Soil-emissions | Agriculture_non-combustion |
| Agricultural soils, pigs in pasture | 3D_Soil-emissions | Agriculture_non-combustion |
| Agricultural soils, sheep in pasture | 3D_Soil-emissions | Agriculture_non-combustion |
| Agricultural soils, turkeys in pasture | 3D_Soil-emissions | Agriculture_non-combustion |
| Indirect N2O emissions | 3D_Soil-emissions | Agriculture_non-combustion |
| Indirect N2O emissions - deposition, other | 3D_Soil-emissions | Agriculture_non-combustion |
| Indirect N2O emissions - deposition, agriculture | 3D_Soil-emissions | Agriculture_non-combustion |
| Indirect N2O emissions - leaching and runoff | 3D_Soil-emissions | Agriculture_non-combustion |
| Agricultural soils, CO2 from urea fertilization | 3D_Soil-emissions | Agriculture_non-combustion |
| Agricultural soils, liming | 3D_Soil-emissions | Agriculture_non-combustion |
| Solid waste disposal (landfills) | 5A_Solid-waste-disposal | Waste |
| Industrial waste water | 5D_Wastewater-handling | Waste |
| Domestic waste water | 5D_Wastewater-handling | Waste |
| Human Waste (not in EDGAR) | 5D_Wastewater-handling | Waste |
| Solid waste disposal (incineration) | 5C_Waste-combustion | Waste |
| Residential waste combustion (not in EDGAR) | 5C_Waste-combustion | Waste |
| Other waste handling | 5E_Other-waste-handling | Waste |
| Coal fires underground | 7A_Fossil-fuel-fires | Fosil_Fuel_Files |

| | | |
|---|---|---|
| Oil fires | 7A_Fossil-fuel-fires | Fosil_Fuel_Files |
| Gas fires | 7A_Fossil-fuel-fires | Fosil_Fuel_Files |

\* This sector is currently equal to zero in all years and countries, and not included in data files.

## A4 Fuel mapping to IEA products


| CEDS Fuel | IEA Product | |
|---|---|---|
| biomass | Industrial waste (TJ-net)<br>Municipal waste (renewable) (TJ-net)<br>Municipal waste (non-renewable) (TJ-net) | Primary solid biofuels (TJ-net)<br>Non-specified primary biofuels/waste (TJ-net)<br>Charcoal (kt) |
| brown_coal | Brown coal (if no detail) (kt)<br>Lignite (kt) | Peat (kt)<br>Peat products (kt) |
| coal_coke | Coke oven coke (kt) | |
| hard_coal | Hard coal (if no detail) (kt)<br>Anthracite (kt)<br>Coking coal (kt)<br>Other bituminous coal (kt)<br>Sub-bituminous coal (kt) | Patent fuel (kt)<br>Gas coke (kt)<br>Coal tar (kt)<br>BKB (kt) |
| light_oil | Refinery feedstocks (kt)<br>Additives/blending components (kt)<br>Other hydrocarbons (kt)<br>Ethane (kt)<br>Liquefied petroleum gases (LPG) (kt)<br>Motor gasoline excl. biofuels (kt)<br>Aviation gasoline (kt)<br>Gasoline type jet fuel (kt) | Kerosene type jet fuel excl. biofuels (kt)<br>Other kerosene (kt)<br>Other Kerosene (kt)<br>Naphtha (kt)<br>White spirit & SBP (kt)<br>Biogasoline (kt)<br>Other liquid biofuels (kt) |
| diesel_oil | Natural gas liquids (kt)<br>Gas/diesel oil excl. biofuels (kt) | Lubricants (kt)<br>Biodiesels (kt) |
| heavy_oil | Oil shale and oil sands (kt)<br>Crude/NGL/feedstocks (if no detail) (kt)<br>Crude oil (kt)<br>Fuel oil (kt) | Bitumen (kt)<br>Paraffin waxes (kt)<br>Petroleum coke (kt)<br>Other oil products (kt) |
| natural_gas | Gas works gas (TJ-gross)<br>Coke oven gas (TJ-gross)<br>Blast furnace gas (TJ-gross)<br>Other recovered gases (TJ-gross) | Natural gas (TJ-gross)<br>Natural Gas (TJ-gross)<br>Refinery gas (kt)<br>Biogases (TJ-net) |
| NOT MAPPED | Elec/heat output from non-specified manufactured gases<br>Heat output from non-specified combustible fuels<br>Nuclear<br>Hydro<br>Geothermal (direct use in TJ-net)<br>Solar photovoltaics<br>Solar thermal (direct use in TJ-net)<br>Tide, wave and ocean | Wind<br>Other sources<br>Electricity (GWh)<br>Heat (TJ)<br>Total<br>Total of all energy sources<br>Memo: Renewables<br>Heat from chemical sources<br>Electric boilers<br>Heat pumps |

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
