# Peer review of "Historical (1750 - 2014) anthropogenic emissions of reactive gases and aerosols from the Community Emission Data System (CEDS)"

_Geoscientific Model Development, 2017_

## Referee Comment (RC1) · Anonymous Referee #1 · 28 Apr 2017

In this paper, the authors document the methodology used to develop a new historical (1750-2014) short-lived species emissions data set for use by global chemistry-climate models for the upcoming Coupled Model Intercomparison Phase 6 (CMIP6). The paper provides detailed information on input data sets (e.g., emissions factors, activity data, population) and steps applied to generate the new emissions trends in Community Emissions Database System. The authors also compare this new dataset with existing emission inventories to place CEDS emissions in the context of existing data sets.

Emissions inventories provide crucial input data for global chemistry-climate models to simulate the spatial and temporal distributions of short-lived pollutants many of which are also climate forcers. Although gridded emissions inventories existed prior to the

inventory of Lamarque et al (2010), there was a lack of consistency in the use of these products by different global modeling groups participating in multi-model intercomparisons. Development of a global gridded emissions inventory for not just the present conditions but going back in time is a major undertaking. I am sure the global chemistry-climate modeling community would be very appreciative of the service provided by the authors in not only updating the previous extensively used inventory (Lamarque et al., 2010), but also creating a consistent, transparent, and trackable process that can hopefully be sustained going forward. The paper is generally well-written and is appropriate for GMD.

My main comment on the paper is that the authors do not provide any comparisons of the spatial distribution of the gridded emissions against existing gridded products. A panel plot with maps of present day (e.g., 2010) species emissions should be included in the main text. Some discussion of how they compare with the spatial distributions in existing inventories would be helpful.

Below are some specific comments to help improve the paper:

Specific comments:

P2: Suggest arranging the discussion of existing emission inventories chronologically. EDGAR has a long history of developing emissions data set and was available much before the inventory of Lamarque et al (2010) (referred to as L2010 hereafter). In fact, the EDGAR informs the L2010 and the work described in this paper.

L84: What is the time period for the historical data?

L85-L87: The chemical formulas for these species are first used on L55 without defining them. Suggest moving the full names closer to where the formulas are used for the first time in the text.

L102-103: At what point is the seasonality added?

Figure 1, captions: replace "produces" with "products" on P4.

L116: What does "energy balance statistics" mean here?

L117: A quick search on google tells me that there are 196 countries in the world (not considering Taiwan separately would bring the count down to 195). Am I missing something? Could the authors please provide a color coded map of countries considered in the work?

L123: A reference is needed for the IPCC guidelines and Nomenclature for reporting document.

L137: What kind of "additional effort" would be needed? Please elaborate.

L140-141: Please elaborate on the "confidentiality issues." As I understand, sector level emissions are provided in the gridded files, so I am confused by this statement.

P4L143: What is meant by "emission control degradation?"

L161-162: Does the population data used to disaggregate energy data for CEDS countries change with time? Describe any assumptions made. Also, please provide a reference for the population data set used.

L170: Is the BP data freely available?

L180: Please provide a reference for the MEIC inventory.

L195-196: This statement conveys ambiguity in the use of population data for generating emission trends. Please clarify.

L214: I feel that this equation can be moved up near the beginning of section 2.1 as it describes clearly how activity data and emissions factors are combined to obtain emissions.

P9L310-311: To clarify, is the "value" of the scaling factor limited to greater than 1/100 and less than 100? If so, please rephrase the sentence.

L344: Replace does with do.

L396-399: Specify that the discussion in paragraph is pertinent to soil NH3 and NOx emissions.

L413: Which CEDS sectors are the authors referring to here? The 55 working sectors, the 16 intermediate sector or the 9 aggregated sectors?

L414: Please clarify what sector (of the 9 aggregated) is the flaring emission relevant for.

P15, Figure 2: The figure caption says that aviation emissions are not included but the color bar shows "Air" as an option. Please clarify

P16, Figure 3: The label "International" to describe international aviation and shipping is misleading. Suggest replacing it with Air_ship (or some combination of air and ship) so that it is clear the authors are referring to combined aviation and shipping emissions.

L451: Please clarify "anthropogenic emissions" from which inventory are being referred to here. Are the CEDS anthropogenic emissions 20-30% of the total global emissions for BC, OC, NMVOC, and CO?

L459: replace "in 1950" with "post 1950"

L460: Insert a reference to Figure 3 at the end of the sentence.

L480: Can the authors postulate any specific reasons for the flat residential biomass emissions in latin America despite a growing rural population, and flatter China emissions than rural population after 1990?

L496: replace 'species of emissions" with "species emissions"

L506: Please refer to a figure to support the statement "Global CO emissions flatten".

L514-515: Please clarify the sentence: "offset by international shipping emissions grow then decrease..."

L516: Is it possible that the decline in North American NOx emissions is driven by

the decreases in US NOx emissions in response to the NOx control regulations implemented in the US (NOx SIP call). This is fairly well documented and literature should be referenced here as this lends confidence to the trends in NOx emissions derived from CEDS.

L521: A reference is needed for "more stringent emission standards for power plants"

L554-L559: Please refer to a specific figure in the Figures and Tables Supplement for this comparison (e.g., Figure S40). I would also suggest doing the same for other species in the paragraphs below as it is cumbersome to sift through the many plots.

Section 3.4: It would be very helpful to have a table with published level of uncertainties in emissions for each species (CO2, SO2, CO, NOx, NMVOC, BC, OC) and specific sectors. Much of the information is contained in this section and can be pulled into a summary table that will come in handy when uncertainties in CEDS emissions are determined.

L622: This sentence can be rephrased to " Emissions uncertainties for CO, NOx, and NMVOCs typically lie between those of carbonaceous aerosols. . .."

L626-627: Hassler et al (2016) should also be cited here.

L637-639: Paulot et al can be cited here as an example of detailed modeling of agricultural NH3 emissions.

Section 5. The ultimate test of an emission inventory is comparison of species concentrations simulated by a model driven by an inventory against observations (e.g., Parrish et al., 2014; Hassler et al., 2016). If the model is able to capture the distribution and trends then the said inventory is considered to represent the real conditions well. I think a case could be made for better coordination between modelers and emission inventory developers so that a two-way interaction can help improve both models and emissions inventories.

An outline of long-term plans for the CEDS database is needed in the summary section

to build confidence in its sustainability. Modelers would like to know if they can rely on the CEDS system working even after CMIP6. What are the plans for maintenance of the back-end software, frequency of updates to the input data and for maintaining funding for CEDS?

References:

Paulot, F., D. J. Jacob, R. W. Pinder, J. O. Bash, K. Travis, and D. K. Henze (2014), Ammonia emissions in the United States, European Union, and China derived by high-resolution inversion of ammonium wet deposition data: Interpretation with a new agricultural emissions inventory (MASAGE_NH3), J. Geophys. Res. Atmos., 119, 4343–4364, doi:10.1002/2013JD021130.

Hassler, B., et al. (2016), Analysis of long-term observations of NOx and CO in megacities and application to constraining emissions inventories, Geophys. Res. Lett., 43, 9920–9930, doi:10.1002/2016GL069894.

---

## Short Comment (SC1) · 5 Jun 2017

I have two very simple questions:

* Where has CH4 gone? It is mentioned in the abstract, but nothing afterwards. It is not shown in any figure, nor is it provided in the supplementary data.

* Where is N2O? It is not mentioned at all.

There may be good reasons for CH4 and N2O to be missing, but these reasons should be at the very least explained. (It looks like CH4 may become available at some point, but it's not very clear for someone who is not closely involved in CMIP6.) If the data

can be found elsewhere, a reference should be given.

Otherwise, recommendation as to what data-sets could complement this one if one wanted to run a global simulation with consistent emission data would be extremely appreciated! (One typical issue, when using older data-sets, is how to extend these over the recent past.)

Even better, this whole work should be done for methane and nitrous oxide, even if it requires more simplistic assumptions. I'm perfectly aware it is no small endeavor, and it is obvious a lot of work has been put into the current version of the data-set. But even if most (all?) complex ESMs do not have interactive CH4- and N2O-cycling, some less complex models do, and the extra data would be very valuable! Ultimately, if CH4 and/or N2O are not required for CMIP6, I do hope another paper for these two gases is on its way...

Mention and recommendation as to the F-gases would be the icing on the cake. (With this same issue of having emission data up to 2014...)

---

## Short Comment (SC2) · 16 Jun 2017

This paper documents input datasets to CMIP6 simulations, and as such is an essential part of the entire CMIP6 process. Below are some comments to help improve the paper.

- insist more on the comparison with the Lamarque 2010 paper, official CMIP5 dataset. Please include spatial comparisons between CMIP5 and CMIP6 total emissions by species, possibly for a selection of years. These maps put light in the discussion. Figures of differences would also be informative.

[Figure]

- although global estimates of SO2 for CEDS and CMIP5 are very similar, spatial plots of the differences (see figure attached untitled DIFFERENCE CMIP6-CMIP5) reveal interesting features (large homogeneous or patchy distributions). Whether in the end you include or not such spatial plots in your paper, please include comments on these differences.

- in Table 2, it is not clear what "(SI-Text 6.3" (and similar references) point to. Please include clarification information.

- in para 3.3 please refer to the figures and/or the tables explicitly.

- please report in a summary Table in this article on the known uncertainty level of emissions, at least by species for a selection of years.
* * *
[Figure]

Fig. 1.

[Figure]

---

## Referee Comment (RC2) · A. Sellar (Referee) · 5 Jul 2017

General comments:

Firstly I congratulate the CEDS team on what has clearly been a monumental effort, and one which will benefit the climate modelling community greatly. On this note, it would be nice to say a little more about the CEDS project itself in the introduction: e.g. what is the "community" aspect, what is the formal project goal.

The manuscript gives a detailed and thorough account of the methodology. Some of the detail could perhaps be moved to the supplementary material to reduce the length

of section 2. This is merely a suggestion: I leave this at the authors' discretion.

Please include some comparison of the spatial distribution of emissions against Lamarque et al. in section 3, at least for the species totals, focussing on 1850 and 2014 since these are important years for CMIP6.

The discussion on uncertainty is very useful, and I am pleased to see that "quantitative uncertainty analysis" and emissions ensembles will be included with future data releases. This will assist in understanding sources of uncertainty in historical radiative forcing due to composition changes; given that this is a major outstanding question in climate modelling I would urge the CEDS project to to place high priority on this development

Finally, given the problems in the 2016 data release which emerged after this discussion paper was published, the manuscript should be updated to refer to the methodological changes which have been applied for the 2017 release, and to summarise the impact of these changes.

Specific comments and suggestions:

P2 L53: "(sometimes also as RCP historical data)" this is not a particularly meaningful phrase. I would suggest using only the name "CMIP5 dataset" for the collective historical and future dataset.

P2 L83: "Preindustrial data (CEDS-v2016-06-18), 1750 – 1850, were released in June 2016 and CMIP6 historical data". Insert date range 1850-2014 for historical data.

P3 L97: This list of 6 phases would be clearer as numbered bullet points (i.e. an {enumerate} environment in LaTeX).

P4 L113: This seems like a key methodological difference from Larmarque et al: it would be good to say something about the impact this difference has on the resulting dataset.

P5 L160: Suggest "available" -> "documented" or "detailed"

P6 L180: "Several other changes were made, such as". Clearly this paper cannot list all such changes, but from a methodological perspective, where is the full set of changes documented? In the CEDS code, or accompanying documentation?

P10 L345: "with a time and sector specific options ...". Delete "a".

P12 L396: "Emissions from mineral and manure emissions are often inconsistently reported; 3B_Manuremanagement and 3D_Soil-emissions together, so CEDS total estimates should be reliable". I think some text is missing here.

P13 L408: "Gridded emissions are aggregated to 9 sectors for final distribution". Does "final distribution" refer to the temporal distribution described in the next sentence? Please make this clearer. Also, why use the intermediate sectors for spatial distribution and the 9 sectors for temporal distribution?

P14 L424: This seems to be repeating what was said on P13 L 411.

P18 L522: missing "due" after comma?

P23 L690: "In future versions of CEDS, quantitative uncertainty analysis will be included for all time periods, but is not complete as of the CMIP6 data version." Does this mean that there is partial uncertainty information in the CMIP6 data version, or none because you will wait for complete information before publishing any? If the former, please say something about the quantitative methodology.

P23 L694: "emissions concentrations are observed". Would "near-source concentrations" be a more accurate description?

---

## Author Comment (AC1) · 6 Jul 2017

**Methane Emissions**

CH$_4$ emissions were released in May 2017 (CEDS v_05_18_2017) as gridded, annual estimates from 1970 – 2014, which are available on Earth System Grid Federation with other CEDS data. Emissions by country and sector will be released with the final version of this manuscript (and are also available on request).

**Data sources and methodology**

Default CH$_4$ emission factors for combustion sectors are calculated similarly to those for NO$_X$, CO, and NH$_3$ emissions, from the global implementation of the GAINS model (Klimont et al., 2016, 2017; Stohl et al., 2015). Default emissions for agriculture sectors (3B_Manure-management, 3D_Rice-Cultivation, and 3D_Rice-Cultivation) are taken from the UN Food and Agriculture Organization data base (FAOSTAT) (FAO, 2016), which are available from 1961 – 2014. Default fugitive petroleum and gas emissions are taken from EDGAR 4.2 emissions (EC-JRC/PBL, 2016) combined with ECLIPSE V5a (Stohl et al., 2015). Remaining non-combustion emissions are taken from EDGAR 4.2 (EC-JRC/PBL, 2016).

The default CH$_4$ emissions estimates are scaled to match to the following inventories: UNFCCC submissions for Annex I countries and the US GHG inventory (US EPA 2016) for the United States. Final methane emissions are just slightly lower than EDGAR estimates, shown in Figure 1.

[Figure]

**Figure 1 Like with like comparison of global CEDS emissions with EDGAR 4.2 , GAINS, and RCP for methane emissions.**

**Comparisons to Other Inventories**

Globally, CEDS emissions range from 93% of RCP values in 1970 to 109% of RCP values in 2000 (Figure 1, Figure x). The CEDS values change more smoothly over time, without a dip in 2000. Because of our use of EDGAR and FAO defaults for most countries, overall CEDS CH4 emissions follow EDGAR values. CEDS energy emissions are consistently larger (22 – 58%) than RCP emissions. CEDS agriculture emissions are consistently 10-15% smaller than RCP estimates, except in 2000 (6% smaller) when RCP estimates dip an CEDS emissions flatten, due to our inclusion of FAO agriculture data.

Global CEDS emissions estimates are slightly smaller than to EDGAR 4.2 estimates, ranging from 94 – 98% the value EDGAR estimates. The similarity is because much of our methane emissions are either from EDGAR,

or FAO (which uses similar methodologies) (Figures 1, 5). The largest differences can be found in 1B2 (fugitive petroleum and gas emissions) in Central and South America Africa, and the Former Soviet Union, as these default emissions also incorporate data from ECLIPSE and 3D (rice cultivation) in China, which is from FAO.

Methane from fugitive oil and gas is a third higher than the value from Larsen et al. (2015), but 12% smaller than Höglund-Isaksson (2015) (Figure 6). Overall energy sector production emissions are almost identical to Höglund-Isaksson, but 33% smaller than the global EPA estimates (comparing CEDS data with the available historical years for each of these data sets). These differences indicate the large uncertainty in fugitive methane emissions from fossil energy production and distribution.

[Figure]

**Figure 2 CEDS methane emissions estimates by aggregate sector, region, and fuel compared to Lamarque et al. (2010). For a like with like comparison, these figures do not include aviation or agricultural waste burning on fields. 'RCO' stands for residential, commercial, and other.**

[Figure]

**Figure 3 Comparison of RCP and CEDS methane estimates globally (top) and by aggregate sector as defined by the RCP data (bottom). For a like with like comparison, these figures do not include aviation or agricultural waste burning on fields. Sectors shown include agriculture, domestic (residential and commercial), energy, industrial, solvents, transportation, and waste.**

[Figure]

**Figure 4 CEDS compared to EDGAR emissions for select diverging sectors.**

**Supplementary CH4 Emissions Extension to 1850**

The May 2017 CEDS release for CH$_4$ emissions only extend from 1970 – 2014 because of additional data needs for consistently estimating emissions for earlier years.

Because a few modeling groups have requested emissions back to 1850, so we will also produce a "rough cut" supplementary extension of CH$_4$ emissions back to 1850 by scaling with RCP/CMIP5 historical CH$_4$ estimates (Lamarque et al. 2010). These pre-1970 estimates were generated by scaling the CEDS 1970 estimates back by aggregate sector at the 26 sub-region level of the RCP/CMIP5 data from Lamarque et al. 2010. While these emission estimates are not fully consistent with the other CEDS emissions, they provide a longer time series, albeit with some additional uncertainty, for groups that would like to have this data.

Note that we have already identified some potential biases in this extended dataset. The waste sector is 30% of total anthropogenic CH4 emissions by 1850. This appears to be due to scaling in the earlier data back in time by population. This is an overestimate of anthropogenic CH4 emissions from this source at that time since landfills and wastewater treatment plants, which create the anaerobic conditions conducive to CH4 emissions, did not start to come into widespread use until around 1930. However, as noted in the main paper, the earlier CMIP5/RCP emission estimates did not distinguish between biomass and coal combustion. Methane emissions from biomass combustion are much larger than those from coal combustion, which means that methane emissions from the residential sector are underestimated in this extrapolation. A rough estimate indicates that these two effects are of similar (and offsetting) magnitude. Further work is necessary to better refine historical CH4 emissions.

**Additional Figures**

[Figure]

**Figure 5 Methane Comparison of CEDS versus RCP by Region**

[Figure]

**Figure 6 Comparison of global Methane from Energy Production Sectors compared to other additional inventories**

**Citations**

EC-JRC/PBL: Emission Database for Global Atmospheric Research (EDGAR), release version 4.3.1. [online] Available from: http://edgar.jrc.ec.europa.eu/overview.php?v=431, 2016.

FAO: FAOSTAT database, [online] Available from: http://www.fao.org/faostat/en/#data, 2016.

Höglund-Isaksson, L.: Global anthropogenic methane emissions 2005–2030: technical mitigation potentials and costs, Atmos. Chem. Phys., 12, 9079-9096, doi:10.5194/acp-12-9079-2012, 2012.

Klimont, Z., Kupiainen, K., Heyes, C., Purohit, P., Cofala, J., Rafaj, P., Borken-Kleefeld, J. and Schöpp, W.: Global anthropogenic emissions of particulate matter including black carbon, Atmospheric Chem. Phys. Discuss., 1–72, doi:10.5194/acp-2016-880, 2016.

Klimont, Z., Hoglund-Isaksson, L., Heyes, C., Rafaj, P., Schopp, W., Cofala, J., Purohit, P., Borken-Kleefeld, J., Kupiainen, K., Kiesewetter, G., Winiwarter, W., Amann, M., Zhao, B., Wang, S., Bertok, I. and Sander, R.: Global scenarios of air pollutants and methane: 1990-2050, ACP (In preparation), 2017.

Lamarque, J.-F., Bond, T. C., Eyring, V., Granier, C., Heil, A., Klimont, Z., Lee, D., Liousse, C., Mieville, A., Owen, B., Schultz, M. G., Shindell, D., Smith, S. J., Stehfest, E., Van Aardenne, J., Cooper, O. R., Kainuma, M., Mahowald, N., McConnell, J. R., Naik, V., Riahi, K. and van Vuuren, D. P.: Historical (1850–2000) gridded anthropogenic and biomass burning emissions of reactive gases and aerosols: methodology and application, Atmos Chem Phys, 10(15), 7017–7039, doi:10.5194/acp-10-7017-2010, 2010.

Larsen, K; Delgado, M; Marsters, P: Untapped Potential: Global Methane Emissions from Oil and Natural Gas Systems. Rhodium Group, 2015. [online] Available from: rhg.com/wp-content/uploads/2015/04/RHG_UntappedPotential_April2015.pdf

Stohl, A., Aamaas, B., Amann, M., Baker, L. H., Bellouin, N., Berntsen, T. K., Boucher, O., Cherian, R., Collins, W., Daskalakis, N., Dusinska, M., Eckhardt, S., Fuglestvedt, J. S., Harju, M., Heyes, C., Hodnebrog, ø., Hao, J., Im, U., Kanakidou, M., Klimont, Z., Kupiainen, K., Law, K. S., Lund, M. T., Maas, R., MacIntosh, C. R., Myhre, G., Myriokefalitakis, S., Olivié, D., Quaas, J., Quennehen, B., Raut, J.-C., Rumbold, S. T., Samset, B. H., Schulz, M., Seland, ø., Shine, K. P., Skeie, R. B., Wang, S., Yttri, K. E. and Zhu, T.: Evaluating the climate and air quality impacts of short-lived pollutants, Atmospheric Chem. Phys., 15(18), 10529–10566, doi:10.5194/acp-15-10529-2015, 2015.

US EPA: EPA report: EPA 430-R-16-002: Inventory of U.S. Greenhouse Gas Emissions and Sinks 1990-2014, U.S. Environmental Protection Agency, 200 Pennsylvania Ave., N.W. Washington, DC 20460, U.S.A., 2012.

---

## Author Comment (AC2) · 1 Sep 2017

Response to Anonymous Referee #1

*In this paper, the authors document the methodology used to develop a new historical (1750-2014) short-lived species emissions data set for use by global chemistry-climate models for the upcoming Coupled Model Intercomparison Phase 6 (CMIP6). The paper provides detailed information on input data sets (e.g., emissions factors, activity data, population) and steps applied to generate the new emissions trends in Community Emissions Database System. The authors also compare this new dataset with existing emission inventories to place CEDS emissions in the context of existing data sets.*

*Emissions inventories provide crucial input data for global chemistry-climate models to simulate the spatial and temporal distributions of short-lived pollutants many of which are also climate forcers. Although gridded emissions inventories existed prior to the inventory of Lamarque et al (2010), there was a lack of consistency in the use of these products by different global modeling groups participating in multi-model intercomparisons. Development of a global gridded emissions inventory for not just the present conditions but going back in time is a major undertaking. I am sure the global chemistry- climate modeling community would be very appreciative of the service provided by the authors in not only updating the previous extensively used inventory (Lamarque et al., 2010), but also creating a consistent, transparent, and trackable process that can hopefully be sustained going forward. The paper is generally well-written and is appropriate for GMD.*

*My main comment on the paper is that the authors do not provide any comparisons of the spatial distribution of the gridded emissions against existing gridded products. A panel plot with maps of present day (e.g., 2010) species emissions should be included in the main text. Some discussion of how they compare with the spatial distributions in existing inventories would be helpful.*

Thank you for your kind comments. A new section has been added to Section 3, titled "Gridded Emissions" which includes a panel figure of gridded emission of CEDS total emissions estimates for all 9 emission species in 2010.

Additionally a discussion of the differences between CEDS grids and Lamarqe et al (2010) grids for 1850 and 2000 have been added to CMIP5 Comparison section. A figure showing side by side gridded maps of the differences for EM for 1850 and 2000 has been added to the main text. Difference maps for all other emissions species for both 1850 and 2000 have been included in the supplemental figures document.

*Below are some specific comments to help improve the paper:*

*Specific comments:*

*P2: Suggest arranging the discussion of existing emission inventories chronologically. EDGAR has a long history of developing emissions data set and was available much before the inventory of Lamarque et al (2010) (referred to as L2010 hereafter). In fact, the EDGAR informs the L2010 and the work described in this paper.*

Thank you for this comment. Paragraphs discussing EDGAR and Lamarque 2010 data (L49 - 76) have been swapped so that this section discusses EDGAR and Gains, followed by Lamarque 2010 data.

*L84: What is the time period for the historical data?*

We have added a section in the appendix, A2.2 that explains versions of CEDS releases. The following text appears in the appendix of the manuscript:

> There have been several releases of the CEDS gridded data. The underlying emissions by country, sector and fuel have been identical in all of these releases, as are total emissions by country and gridding sector (with the exception of small changes in 1850 emissions noted below).
>
> **v2016-05-20**: Pre-industrial 1750-1850 data release
>
> **v2016-06-18**: 1851 – 2014 data
>
> **v2016-06-18-sectorDim**: Re-release of both preindustrial and 1851 – 2014 in a new netCDF format with sectors as an additional dimension in the data variable. This reformatting was necessary due to a limitation that was discovered within the ESGF system summer 2016. The reformatted data were released early Fall 2016
>
> **2017-05-18**: Re-release of entire dataset in order to correct two gridding errors discovered by users. 1) Inconsistent emission allocation to spatial grids within countries that resulted in incorrect spatial allocations and some large discontinuities in the gridded data. These issues were particularly apparent in spatially large countries such as the USA and China. 2) Minor inconsistencies in seasonal allocation, resulting largely in emissions that were too high in February. Total annual emissions within each country were not impacted by either of these issues.
>
> Emissions are also fully consistent across 1850 in this release. There were small discontinuities in 1850 between the CEDS CMIP6 preindustrial release (v2016-06-18) and the later full CEDS release (v2016-07-26) due to updates in the data system. These differences are 0.5% for all species (except NMVOC which reaches 1.5%). In absolute terms these differences are very small (relative to, for example, open biomass burning emissions) and will not have a significant impact on simulation results.
>
> A link to further examination of these issues, including comparison maps and time series comparisons, can be found at the project web site (globalchange.umd.edu/CEDS).

*L85-L87: The chemical formulas for these species are first used on L55 without defining them. Suggest moving the full names closer to where the formulas are used for the first time in the text.*

Thank you for pointing this out, Appropriate explanations of chemical formulas were moved to L55 where they first appear.

*L102-103: At what point is the seasonality added?*

The following text was changed to specify that seasonality is added to gridded data in the final step: "…6) gridded emissions with monthly seasonality are produced from aggregate estimates using proxy data.." There is additional explanation of this process in section 2.6 Gridded Emissions.

*Figure 1, captions: replace "produces" with "products" on P4.*

Thank you. This change was made

*L116: What does "energy balance statistics" mean here?*

We are refereeing to detailed IEA energy statistics. The following change has been added for clarification:

"energy statistics"

*L117: A quick search on google tells me that there are 196 countries in the world (not considering Taiwan separately would bring the count down to 195). Am I missing some- thing? Could the authors please provide a color coded map of countries considered in the work?*

We clarify in the revised manuscript that we consider a number of regions whose exact status might not be clear. The most definitive categorization is given by the UN, and we use UN population data as the basis of our current "country" disaggregation. As noted in the manuscript, we are using the term "country" regardless of the exact status of any particular entity.

The supplemental information includes a csv file which contains a list of all the "countries" used here along with their common name, ISO code, and mapping to countries and regions from other data sets such as the IEA energy data. Many of these "countries" would not be visible on a global color coded map as they are small islands and territories.

The following change has been added to the manuscript for clarification:

"CEDS estimates emissions for 221 **regions** (and a global region for international shipping and aircraft), ... **"Regions" refers to countries, regions, territories, or islands and are listed, along with mapping to regions and ISO codes in the supplemental files; they will henceforth be referred to as "countries"."**

*L123: A reference is needed for the IPCC guidelines and Nomenclature for reporting document.*

This reference has been added. Thank you.

*L137: What kind of "additional effort" would be needed? Please elaborate.*

These efforts are briefly described in the future work section of the manuscript. The following text has been added to the manuscript to clarify:

Greater disaggregation for these sectors would improve these estimates, but will require additional effort**, described in Sect.5 Limitations and Future work.**

*L140-141: Please elaborate on the "confidentiality issues." As I understand, sector level emissions are provided in the gridded files, so I am confused by this statement.*

The text has been clarified on this point. We note that other global emissions data providers, such as EDGAR, are subject to a similar limitation in terms of releasing emissions data at the level of fuel, sector, and country.

The core outputs of the CEDS system are country-level emissions aggregated to the CEDS sector level. Emissions by fuel and **detailed CEDS** sector are also available within the system for analysis, although these are not released because this could violate the terms of our use of the IEA energy statistics. (This is the same reason other global inventory data, such as EDGAR, also do not release data by sector, country, and fuel). Emissions are further aggregated and processed to provide gridded emissions data with monthly seasonality, detailed in Sect. …

*P4L143: What is meant by "emission control degradation?"*

"Emission control degradation" refers to certain emission control technologies that may become less effective overtime, for example an old catalytic converter in an old car may be less effective than a new catalytic in an old car. The following text has been changed in the manuscript for clarification:

For example, CEDS does not include a representation of vehicle fleet turnover and emission control degradation (*e.g.* **the effectiveness of catalytic converters over time**) or multiple fuel combustion technologies that are included in more detailed inventories.

*L161-162: Does the population data used to disaggregate energy data for CEDS countries change with time? Describe any assumptions made. Also, please provide a reference for the population data set used.*

As described in section 2.2.2 and further the supplement, a full time series over 1750-2014 of population estimates for all CEDS "countries" is developed by merging several data sources.

However, this explanation incorrectly describes the methodology we used for disaggregating IEA aggregate regions. Aggregate IEA data were disaggregated using CDIAC data, not population data. The following text now appears in the manuscript:

Data for a number of small countries provided by IEA only at an aggregate level, such as "Other Africa" and "Other Asia", are disaggregated to CEDS countries using historical $CO_2$ emissions data from the Carbon Dioxide Information Analysis Center (CDIAC) (Andres et al., 2012; Boden et al., 1995).

*L170: Is the BP data freely available?*

Yes. BP data is publically available online. The following text has been changed for clarification

IEA energy statistics were extended to 2014 using BP Statistical Review of World Energy (BP,

2015), **which is freely available online** and provides annual updates of country energy

*L180: Please provide a reference for the MEIC inventory.*

The MEIC citation has been added. Thank you.

*L195-196: This statement conveys ambiguity in the use of population data for generating emission trends. Please clarify.*

This text has been changed to:

"While non-combustion emissions use population as an "activity driver" in calculations, emissions trends are determined by a combination of EDGAR and country level inventories. Final emissions estimates, therefore, reflect recent emissions inventories where these are available, rather than population trends."

*L214: I feel that this equation can be moved up near the beginning of section 2.1 as it describes clearly how activity data and emissions factors are combined to obtain emissions.*

Thank you for your suggestion. We've kept the formal equation where it is to avoid restructuring a too much text, but added the simple "emissions = driver x emission factor" phrase to section 2.1.

*P9L310-311: To clarify, is the "value" of the scaling factor limited to greater than 1/100 and less than 100? If so, please rephrase the sentence.*

Yes, that is correct. Thank you for this comment. The following change has been made in the manuscript:

Calculated scaling factors are **limited to values between 1/100 and 100. Scaling factors outside this range** may result from…

*L344: Replace does with do.*

This change has been made, thank you.

*L396-399: Specify that the discussion in paragraph is pertinent to soil NH3 and NOx emissions.*

"Emissions from mineral and manure emissions…" has been changed to "$NH_3$ and $NO_X$ emissions from mineral and manure are often…"

*L413: Which CEDS sectors are the authors referring to here? The 55 working sectors, the 16 intermediate sector or the 9 aggregated sectors?*

By "in most sectors" we mean, for most of the data, which could mean any sector aggregation. To avoid confusion "Proxy data used for gridding in most CEDS sectors are primarily gridded emissions …" has been changed to "Proxy data used for gridding are primarily gridded emissions from…" Thank you for this comment

*L414: Please clarify what sector (of the 9 aggregated) is the flaring emission relevant for.*

Flaring emissions are one of the intermediate gridding sectors within the energy sector for final gridding. Final gridding sectors have been added to table 6 for clarification.

*P15, Figure 2: The figure caption says that aviation emissions are not included but the color bar shows "Air" as an option. Please clarify*

Even though "Air" was in the legend, the figure did not show any emissions, as they were not included in the graph. The Air sector has been removed from the legend in these figures for clarity.

*P16, Figure 3: The label "International" to describe international aviation and shipping is misleading. Suggest replacing it with Air_ship (or some combination of air and ship) so that it is clear the authors are referring to combined aviation and shipping emissions.*

The region "International" has been changed to "International Air-Ship" throughout the paper.

*L451: Please clarify "anthropogenic emissions" from which inventory are being referred to here. Are the CEDS anthropogenic emissions 20-30% of the total global emissions for BC, OC, NMVOC, and CO?*

The following paragraph has been rearranged to read:

"In 1850, the earliest year in which most existing data sets provide estimates, anthropogenic emissions are dominated by residential sector cooking and heating and therefore products of incomplete combustion for BC, OC, CO, and NMVOC. In 1850, anthropogenic emissions (sectors included in this inventory), make up approximately 20 – 30% of total global emissions (which also include grassland and forest burning, estimated by Lamarque et al. (2010)) for BC, OC, NMVOC, and CO but only 3% of global $NO_X$ emissions."

*L459: replace "in 1950" with "post 1950"*

This change has been made.

*L460: Insert a reference to Figure 3 at the end of the sentence.*

This change has been made.

*L480: Can the authors postulate any specific reasons for the flat residential biomass emissions in latin America despite a growing rural population, and flatter China emissions than rural population after 1990?*

The following text was added:

"While rural population in China continually grows, residential biomass use flattens in 1990 as both the share of urban population in China increases and rural residential per capita biomass use

decreases."

*L496: replace 'species of emissions" with "species emissions"*

This change has been made

*L506: Please refer to a figure to support the statement "Global CO emissions flatten".*

The following text has been added to the sentence: "… shown in Figure 2 and in more detail in the Supplemental figures and tables."

*L514-515: Please clarify the sentence: "offset by international shipping emissions grow then decrease…"*

Thank you for pointing out this poorly phrased sentence. This sentence has been changed to:

"Global $NO_X$ emissions rise then flatten around 2008. The growth in industrial emissions after 2000 is offset in 2007 by the decrease in international shipping emissions, while global emissions in other sectors stay flat."

*L516: Is it possible that the decline in North American NOx emissions is driven by the decreases in US NOx emissions in response to the NOx control regulations implemented in the US (NOx SIP call). This is fairly well documented and literature should be referenced here as this lends confidence to the trends in NOx emissions derived from CEDS.*

Thank you. A reference was added.

*L521: A reference is needed for "more stringent emission standards for power plants"*

Thank you. A reference was added.

*L554-L559: Please refer to a specific figure in the Figures and Tables Supplement for this comparison (e.g., Figure S40). I would also suggest doing the same for other species in the paragraphs below as it is cumbersome to sift through the many plots.*

Thanks. Specific references to supplemental figures have been added strategically in the results sections of the manuscript.

*Section 3.4: It would be very helpful to have a table with published level of uncertainties in emissions for each species (CO2, SO2, CO, NOx, NMVOC, BC, OC) and specific sectors. Much of the information is contained in this section and can be pulled into a summary table that will come in handy when uncertainties in CEDS emissions are determined.*

While we agree this would be useful, we will refer the reader to the literature summary in IPCC AR5 for now (we're not aware of a more up to date general summary) as more significant effort to collect uncertainty estimates will need to wait for future work.

*L622: This sentence can be rephrased to " Emissions uncertainties for CO, NOx, and NMVOCs*

*typically lie between those of carbonaceous aerosols....”*

This change has been made. Thanks.

*L626-627: Hassler et al (2016) should also be cited here.*

This citation has been added.

*L637-639: Paulot et al can be cited here as an example of detailed modeling of agricultural NH3 emissions.*

This citation has been added.

*Section 5. The ultimate test of an emission inventory is comparison of species concentrations simulated by a model driven by an inventory against observations (e.g., Parrish et al., 2014; Hassler et al., 2016). If the model is able to capture the distribution and trends then the said inventory is considered to represent the real conditions well. I think a case could be made for better coordination between modelers and emission inventory developers so that a two-way interaction can help improve both models and emissions inventories.*

We agree in general, although note that models are not necessary in all cases: Hassler et al., for example, use observations directly. There are many complications of course: models are imperfect or incomplete, observations are often not available at the same scale as model results, and inventories are often not available at scale of observations (e.g. Wang et al. doi/10.1073/pnas.1318763111 ).

We have added a comment on this in the future work section.

*An outline of long-term plans for the CEDS database is needed in the summary section to build confidence in its sustainability. Modelers would like to know if they can rely on the CEDS system working even after CMIP6. What are the plans for maintenance of the back-end software, frequency of updates to the input data and for maintaining funding for CEDS?*

Thanks. The following text has been added to the manuscript detailing future plans for the System and community engagement.

"The CEDS data system, including R code and all input data other than the IEA energy balances, is being prepared for public release in fall 2017 through the gitHub collaboration website. This will facilitate community comment, and direct contributions to improving these emissions data. The next data release is planned for Fall 2017, which will extend the time series to 2016 and correct, to the extent possible, any known issues with the dataset. We aim to continue annual updates in subsequent years."

References:

Paulot, F., D. J. Jacob, R. W. Pinder, J. O. Bash, K. Travis, and D. K. Henze (2014), Ammonia

emissions in the United States, European Union, and China derived by high- resolution inversion of ammonium wet deposition data: Interpretation with a new agri- cultural emissions inventory (MASAGE_NH3), J. Geophys. Res. Atmos., 119, 4343– 4364, doi:10.1002/2013JD021130.

Hassler, B., et al. (2016), Analysis of long-term observations of NOx and CO in megac- ities and application to constraining emissions inventories, Geophys. Res. Lett., 43, 9920–9930, doi:10.1002/2016GL069894.

---

## Author Comment (AC3) · 1 Sep 2017

Response to **A. Sellar (Referee)**

*General comments:*

*Firstly I congratulate the CEDS team on what has clearly been a monumental effort, and one which will benefit the climate modelling community greatly. On this note, it would be nice to say a little more about the CEDS project itself in the introduction: e.g. what is the "community" aspect, what is the formal project goal.*

Thank you for your kind comments. The following text on "community" has been added to the summary section:

With release of this data set, and soon the entire data system, it is our intention that further improvements will be made through feedback from the global emissions inventory community. The CEDS data system, including R code and all input data other than the IEA energy balances, is being prepared for public release in fall 2017 through the gitHub collaboration website. This will facilitate community comment, and direct contributions to improving these emissions data. The next data release is planned for Fall 2017, which will extend the time series to 2016 and correct, to the extent possible, any known issues with the dataset. We aim to continue annual updates in subsequent years. We welcome comments, including notes on any potential inconstancies or relevant new data sources, so that that these data can be improved in future releases.

*The manuscript gives a detailed and thorough account of the methodology. Some of the detail could perhaps be moved to the supplementary material to reduce the length of section 2. This is merely a suggestion: I leave this at the authors' discretion.*

Thank you for the suggestion. We agree that the paper is a bit long, and we have moved a few secondary points to the supplement, but during our internal review process found that we received many methodology questions, as many readers (unless motivated by a specific highly detailed questions) are unlikely to page through the extensive supplement.

*Please include some comparison of the spatial distribution of emissions against Lamarque et al. in section 3, at least for the species totals, focusing on 1850 and 2014 since these are important years for CMIP6.*

We have added a section showing gridded data, as well as comparisons of gridded data between Lamarque et al 2010 and CEDS data. We have included additional gridded figures in the SI as well.

*The discussion on uncertainty is very useful, and I am pleased to see that "quantitative uncertainty analysis" and emissions ensembles will be included with future data releases. This will assist in understanding sources of uncertainty in historical radiative forcing due to composition changes; given that this is a major outstanding question in climate modelling I would urge the CEDS project to place high priority on this development.*

Thank you. We agree. After the upcoming public release of the data system, uncertainty is a high priority.

*Finally, given the problems in the 2016 data release which emerged after this discussion paper was published, the manuscript should be updated to refer to the methodological changes which have been applied for the 2017 release, and to summarize the impact of these changes.*

We've added a section in the appendix detailing the changes in the various releases of the gridded data.

*Specific comments and suggestions:*

*P2 L53: "(sometimes also as RCP historical data)" this is not a particularly meaningful phrase. I would suggest using only the name "CMIP5 dataset" for the collective historical and future dataset.*

This text has been changed to:
"This data is also used as the historical starting point for the Representative Concentration Pathways (RCP) scenarios (van Vuuren et al., 2011) and in some research communities is referred to as the RCP historical data. In this article it is referred to as the CMIP5 data set."

*P2 L83: "Preindustrial data (CEDS-v2016-06-18), 1750 – 1850, were released in June 2016 and CMIP6 historical data". Insert date range 1850-2014 for historical data.*

We have added a section in the appendix, A2.2 that explains versions of CEDS releases. The following text appears in the appendix of the manuscript:

> There have been several releases of the CEDS gridded data. The underlying emissions by country, sector and fuel have been identical in all of these releases, as are total emissions by country and gridding sector (with the exception of small changes in 1850 emissions noted below).
>
> **v2016-05-20**: Pre-industrial 1750-1850 data release
>
> **v2016-06-18**: 1851 – 2014 data
>
> **v2016-06-18-sectorDim**: Re-release of both preindustrial and 1851 – 2014 in  a new netCDF format with sectors as an additional dimension in the data variable. This reformatting was necessary due to a limitation that was discovered within the ESGF system summer 2016. The reformatted data were released early Fall 2016
>
> **2017-05-18**: Re-release of entire dataset in order to correct two gridding errors discovered by users. 1) Inconsistent emission allocation to spatial grids within countries that resulted in incorrect spatial allocations and some large discontinuities in the gridded data. These issues were particularly apparent in spatially large countries such as the USA and China. 2) Minor inconsistencies in seasonal allocation, resulting largely in emissions that were too high in February. Total annual emissions within each country were not impacted by either of these issues.

Emissions are also fully consistent across 1850 in this release. There were small discontinuities in 1850 between the CEDS CMIP6 preindustrial release (v2016-06-18) and the later full CEDS release (v2016-07-26) due to updates in the data system. These differences are 0.5% for all species (except NMVOC which reaches 1.5%). In absolute terms these differences are very small (relative to, for example, open biomass burning emissions) and will not have a significant impact on simulation results.

A link to further examination of these issues, including comparison maps and time series comparisons, can be found at the project web site (globalchange.umd.edu/CEDS).

*P3 L97: This list of 6 phases would be clearer as numbered bullet points (i.e. an {enumerate} environment in LaTeX).*

This change has been made. Thank you.

*P4 L113: This seems like a key methodological difference from Larmarque et al: it would be good to say something about the impact this difference has on the resulting dataset.*

This is a large change in methodology. First, it allows us to extend estimates forward over recent years by using recently updated energy data. Second it allows us to use more detail in historical years by modeling fuel use and EFs separately, which has had an impact on NOx and CO emissions from residential biomass burning for example. While these impacts are not explicitly noted in this section of the paper, we feel the rest of the paper has highlighted these differences, specifically the section on comparisons to the CMIP5 (*Larmarque et al* ) dataset. A major advantage of this method is also that it can more consistency capture trends over time, mentioned here in the text, and which results in some changes in recent trends discussed later in the comparison with the CMIP5 dataset.

*P5 L160: Suggest "available" -> "documented" or "detailed"*

Thank you. The following text now appears in the manuscript:

Mapping of IEA products to CEDS fuels is **provided** in Sect. A3.

*P6 L180: "Several other changes were made, such as". Clearly this paper cannot list all such changes, but from a methodological perspective, where is the full set of changes documented? In the CEDS code, or accompanying documentation?*

Most, if not all of these assumptions are detailed in the Data and Assumption Supplement. All detailed methods (e.g. code), assumptions, and data will also be available with the open source release of the system. This is detailed in the manuscript in additional text describing the release of the system, as well as existing text describing the substantial supplemental information available to download with the manuscript.

*P10 L345: "with a time and sector specific options ...". Delete "a".*

This change has been made. Thank you

*P12 L396: "Emissions from mineral and manure emissions are often inconsistently reported;*

*3B_Manuremanagement and 3D_Soil-emissions together, so CEDS total estimates should be reliable". I think some text is missing here.*

This has been fixed. Thanks.

*P13 L408: "Gridded emissions are aggregated to 9 sectors for final distribution". Does "final distribution" refer to the temporal distribution described in the next sentence? Please make this clearer. Also, why use the intermediate sectors for spatial distribution and the 9 sectors for temporal distribution?*

Emissions are distributed over space using the intermediate gridding sectors. Seasonality is added using the 9 final gridding sectors. These steps in aggregation are determined by the level of detail of the proxy data and seasonality profiles. We've rearranged and added some language to make this more clear in the manuscript. Thank you.

*P14 L424: This seems to be repeating what was said on P13 L 411.*

Thanks. We've moved some text around so it is less repetitive.

*P18 L522: missing "due" after comma?*

Thanks. This change has been made.

*P23 L690: "In future versions of CEDS, quantitative uncertainty analysis will be included for all time periods, but is not complete as of the CMIP6 data version." Does this mean that there is partial uncertainty information in the CMIP6 data version, or none because you will wait for complete information before publishing any? If the former, please say something about the quantitative methodology.*

There is no quantitative uncertainty analysis at this time other than what is already in the literature. (The text has been revised to refer to an existing literature summary.)

*P23 L694: "emissions concentrations are observed". Would "near-source concentrations" be a more accurate description?*

We would like to note that comparison to both near-source and far field observations can be helpful in comparisons such as these, and we have compared to both in the paper (Hassler et al. 2016 and Kanaya et al. 2016).

---

## Author Response (AR2)

**Topical Editor Decision: Publish subject to technical corrections** (10 Nov 2017) by Fiona O'Connor
Comments to the Author:
Dear Rachel and co-authors,

Firstly, I would like to acknowledge the tremendous effort that has gone into producing this emission dataset and the value of this dataset to the global composition climate community, as indicated by the referees. I also thank them for their effort in reviewing your manuscript and for providing useful and constructive feedback. Many thanks to you for submitting your responses to their comments and for submitting a revised manuscript.

Secondly, I also sincerely apologise for the delay in reaching a decision on your manuscript.

Having read the reviewer comments, your responses to them, and the revisions you've made to the manuscript, I am pleased to inform you that your paper has been accepted for publication in Geosci. Model Development.

However, there are a small number of minor specific comments below which I would like you to implement before final publication.

Congratulations again to you and your co-authors on your efforts,

Regards,
Fiona O'Connor

Minor corrections:
1. Summary, page 35, line 1028: Replace "release of this data" with "the release of this data"
2. Summary, page 35, line 1036: Replace "inconstancies" with "inconsistencies"; Also, remove one of the two "that"
3. Referee (Alistair Sellar) had the following comment:

P2 L53: "(sometimes also as RCP historical data)" this is not a particularly meaningful phrase. I would suggest using only the name "CMIP5 dataset" for the collective historical and future dataset."

to which you replied with:

This text has been changed to: "This data is also used as the historical starting point for the Representative Concentration Pathways (RCP) scenarios (van Vuuren et al., 2011) and in some research communities is referred to as the RCP historical data. In this article it is referred to as the CMIP5 data set."

However, the revised manuscript does not include the full text from your response and I would ask that you correct this – see page 3, lines 93-95 (I think it has been altered in one place but not on page 3, lines 93-95)

4. Referee (Alistair Sellar) has the following specific comment:

P5 L160: Suggest "available" -> "documented" or "detailed"

Your reply was as follows:

Thank you. The following text now appears in the manuscript: Mapping of IEA products to CEDS fuels is provided in Sect.A3.

As above, the revised manuscript did not contain the response as indicated above. Although a minor point, may I please ask you to correct this oversight?

This correction has been made. Thank you for catching this oversight.

5. Referee (Alistair Sellar) made the following comment:

P23 L690: "In future versions of CEDS, quantitative uncertainty analysis will be included for all time periods, but is not complete as of the CMIP6 data version." Does this mean that there is partial uncertainty information in the CMIP6 data version, or none because you will wait for complete information before publishing any? If the former, please say something about the quantitative methodology.

Your response was:

There is no quantitative uncertainty analysis at this time other than what is already in the literature. (The text has been revised to refer to an existing literature summary.)

However, it is unclear to me that the text referred to was modified at all in the revised manuscript. Please clarify.

Thank you for catching this oversight. The following text has been changed in the Uncertainty section:

"In future versions of CEDS, quantitative uncertainty analysis will be included for all time periods, and is further explained in the Sect. 5."

The following text already appears in Section 5:

"A major next step in this project will be estimation of uncertainty. Our first step will be quantification of the additional uncertainty that stems from producing estimates out to the most recent full year, followed by comprehensive uncertainty estimates that will be used to produce ensembles of emissions to more fully reflect the uncertainty in these data."

6. Referee (Alistair Sellar) commented:

P23 L694: "emissions concentrations are observed". Would "near-source concentrations" be a more accurate description?

You replied:

We would like to note that comparison to both near-source and far field observations can be helpful in comparisons such as these, and we have compared to both in the paper (Hassler et al. 2016and Kanaya et al. 2016).

I agree with the value of comparing to near-source and far field observations. However, I think the referee here is simply referring to your use of language in the phrase "emissions concentrations" rather than having a problem with your methodology. Therefore, I would like you to replace the phrase "emissions concentrations" to "concentrations of emitted species" (Pg 31, line 909)

This change has been made. Thank you.

7. Anonymous Referee #1 made the following comment:

L123: A reference is needed for the IPCC guidelines and Nomenclature for reporting document.

Your reply was:

This reference has been added. Thank you.

However, it is not clear that this reference has been added on page 5 line 158 of revised manuscript. Please add as requested by referee.

Thank you for catching this oversight. This reference has been added in text and in the references section as well as the following small footnote:

[1] Sector names were derived NFR14 nomenclature via a mapping table provided by CEIP, available from: http://www.ceip.at/ms/ceip_home1/ceip_home/reporting_instructions/

8. In response to Anonymous Referee #1, you have altered the legend of Figure 2. For consistency, I would suggest that you also alter the legend of other figures (e.g. Figure 3 and 5) as, again, no aircraft emissions are considered.

This change has been made. Thank you.

9. Anonymous Referee #1 asked for a table of uncertainties. While this would be beneficial, my feeling is that it is too big a request for this paper and would be more appropriate when CEDS emission uncertainty estimates have been developed and are being published. I accept your argument on this point.

10. Anonymous Referee #1 made a comment about the potential for two-way interaction

between emission inventory developers and the modelling community. Although your response to this comment indicated that you had added something in the future work section, this wasn't evident in the marked-up manuscript. Can you please a comment to the future work section? Thanks.

That was our mistake in the response document. The following text was added to the Summary section, not the Future work section. We believe this text satisfies the request for a discussion on two-way interaction.

[revised manuscript text omitted]

 Anthracite (kt)
 Coking coal (kt)
 Other bituminous coal (kt)
 Sub-bituminous coal (kt) | Patent fuel (kt)
 Gas coke (kt)
 Coal tar (kt)
 BKB (kt) |
| light_oil | Refinery feedstocks (kt)
 Additives/blending components (kt)
 Other hydrocarbons (kt)
 Ethane (kt)
 Liquefied petroleum gases (LPG) (kt)
 Motor gasoline excl. biofuels (kt)
 Aviation gasoline (kt)
 Gasoline type jet fuel (kt) | Kerosene type jet fuel excl. biofuels (kt)
 Other kerosene (kt)
 Other Kerosene (kt)
 Naphtha (kt)
 White spirit & SBP (kt)
 Biogasoline (kt)
 Other liquid biofuels (kt) |
| diesel_oil | Natural gas liquids (kt)
 Gas/diesel oil excl. biofuels (kt) | Lubricants (kt)
 Biodiesels (kt) |
| heavy_oil | Oil shale and oil sands (kt)
 Crude/NGL/feedstocks (if no detail) (kt)
 Crude oil (kt)
 Fuel oil (kt) | Bitumen (kt)
 Paraffin waxes (kt)
 Petroleum coke (kt)
 Other oil products (kt) |
| natural_gas | Gas works gas (TJ-gross)
 Coke oven gas (TJ-gross)
 Blast furnace gas (TJ-gross)
 Other recovered gases (TJ-gross) | Natural gas (TJ-gross)
 Natural Gas (TJ-gross)
 Refinery gas (kt)
 Biogases (TJ-net) |
| NOT MAPPED | Elec/heat output from non-specified manufactured gases
 Heat output from non-specified combustible fuels
 Nuclear
 Hydro
 Geothermal (direct use in TJ-net)
 Solar photovoltaics
 Solar thermal (direct use in TJ-net)
 Tide, wave and ocean | Wind
 Other sources
 Electricity (GWh)
 Heat (TJ)
 Total
 Total of all energy sources
 Memo: Renewables
 Heat from chemical sources
 Electric boilers

[revised manuscript text omitted]